# ARMH3 is an ARL5 effector that promotes PI4KB-catalyzed PI4P synthesis at the *trans*-Golgi network

Morié Ishida [1,4], Adriana E. Golding [1,4], Tal Keren-Kaplan[1], Yan Li [2], Tamas Balla[3] & Juan S. Bonifacino [1]✉

ARL5 is a member of the ARF family of small GTPases that is recruited to the *trans*-Golgi network (TGN) by another ARF-family member, ARFRP1, in complex with the transmembrane protein SYS1. ARL5 recruits its effector, the multisubunit tethering complex GARP, to promote SNARE-dependent fusion of endosome-derived retrograde transport carriers with the TGN. To further investigate the function of ARL5, we sought to identify additional effectors. Using proximity biotinylation and protein interaction assays, we found that the armadillo-repeat protein ARMH3 (C10orf76) binds to active, but not inactive, ARL5, and that it is recruited to the TGN in a SYS1-ARFRP1-ARL5-dependent manner. Unlike GARP, ARMH3 is not required for the retrograde transport of various cargo proteins. Instead, ARMH3 functions to activate phosphatidylinositol 4-kinase IIIβ (PI4KB), accounting for the main pool of phosphatidylinositol 4-phosphate (PI4P) at the TGN. This function contributes to recruitment of the oncoprotein GOLPH3 and glycan modifications at the TGN. These studies thus identify the SYS1-ARFRP1-ARL5-ARMH3 axis as a regulator of PI4KB-dependent generation of PI4P at the TGN.

The mammalian ADP-ribosylation factor (ARF) family of small GTPases comprises approximately 30 members that act as molecular switches to regulate various intracellular processes, including membrane trafficking, organelle motility, cytoskeletal dynamics, cilia formation, and cell migration[1–3]. Mutations in several members of this family cause various diseases, underscoring their importance in human physiology[2–4]. Like other small GTPases, ARF-family members cycle between GTP-bound, active, and GDP-bound, inactive forms. GTP binding induces a conformational change in the switch I and II regions, exposing a binding site for a diverse set of effector proteins[1–3]. Additionally, a reorganization of the interswitch region pushes a myristoylated or acetylated N-terminal amphipathic α-helix away from the GTPase core and into the lipid bilayer[1–3]. This mechanism thus enables the coupling of effector binding to membrane recruitment.

GTPase activation is mediated by guanine-nucleotide exchange factors (GEFs), which promote the dissociation of GDP and thereby the association of GTP. Conversely, GTPase inactivation is mediated by GTPase-activating proteins (GAPs), which facilitate hydrolysis of the bound GTP to GDP[1–3]. The dynamic interplay between GEFs and GAPs enables GTPase cycling in response to diverse cellular signals.

Some members of this family, like the ARF and SAR1 GTPases (*e.g.*, ARF1, ARF3, ARF4, ARF5, ARF6, SAR1A, and SAR1B in mammals) have been extensively characterized, and multiple effectors, GEFs, GAPs, and functions are known[1–3]. In contrast, other members like the ARL GTPases (of which there are approximately 20 in mammals) are less understood, with limited information available regarding their interactors and functions[1–3].

[1]Division of Neurosciences and Cellular Structure, Eunice Kennedy Shriver National Institute of Child Health and Human Development, National Institutes of Health, Bethesda, MD, USA. [2]Proteomics Core Facility, National Institute of Neurological Disorders and Stroke, National Institutes of Health, Bethesda, MD, USA. [3]Section on Molecular Signal Transduction, Eunice Kennedy Shriver National Institute of Child Health and Human Development, National Institutes of Health, Bethesda, MD, USA. [4]These authors contributed equally: Morié Ishida, Adriana E. Golding. ✉e-mail: juan.bonifacino@nih.gov

Recent studies have begun to shed light on the functions of ARL5, which, in mammals, exists as two closely related paralogs, ARL5A and ARL5B[5,6]. Unless otherwise specified, herein we refer to both paralogs collectively as ARL5 for simplicity. ARL5 is recruited to the *trans*-Golgi network (TGN)[5,6] by the GTP-bound form of another ARL GTPase, ARFRP1, in complex with the multispanning membrane protein SYS1[7]. The GEFs and GAPs that regulate the ARL5 nucleotide cycle, as well as their possible relationship to SYS1-ARFRP1, are currently unknown. At the TGN, GTP-bound ARL5 recruits a multi-subunit tethering complex known as Golgi-associated retrograde protein (GARP), composed of VPS51, VPS52, VPS53, and VPS54 subunits[6,7]. GARP functions to promote the SNARE-mediated fusion of endosome-derived vesicular or tubular carriers with the TGN[8–10]. ARL5 thus contributes to the retrograde transport of various cargos, such as TGN46, the cation-independent mannose 6-phosphate receptor (CI-MPR), and the Shiga toxin B subunit (STxB), from endosomes to the TGN[5–7].

Considering that some of the more extensively characterized members of the ARF family (*e.g.*, ARF1 and ARF3) have over 15 known effectors[1–3], we hypothesized that ARL5 might also have additional effectors. The identification of such effectors is crucial for unraveling the complete range of functions performed by ARL5 at the TGN. Herein, we present the findings of a search for additional effectors of ARL5 using MitoID[11,12], a proximity biotinylation-mass spectrometry method using mitochondrially-targeted forms of ARL5A and ARL5B as baits. This approach identified the protein ARMH3 (also known as C10orf76) as a specific interactor of active, but not inactive, ARL5A and ARL5B. ARMH3 was previously reported to be a regulator of phosphatidylinositol 4-kinase beta (PI4KB, also known as PI4KIIIbeta) at the TGN[13,14]. Yeast two-hybrid (Y2H) assays confirmed the ARL5-ARMH3 interactions, suggesting that they are direct. We also found that ARMH3 localizes to the TGN in a SYS1-ARFRP1-ARL5-dependent manner, enhancing PI4KB-dependent synthesis of PI4P. Local PI4P then promotes the recruitment of the PI4P-binding oncoprotein GOLPH3 to the TGN[15]. Consistent with the previously demonstrated role of GOLPH3 in maintaining the localization and stability of Golgi glycosylation enzymes[16–21], we found that ARMH3 and PI4KB are required for terminal glycan modifications. In contrast to GARP, ARMH3, and PI4KB are dispensable for the retrograde transport of various cargos from endosomes to the TGN. These findings thus identify a SYS1-ARFRP1-ARL5-ARMH3-PI4KB pathway for the regulation of PI4P synthesis at the TGN, and demonstrate that ARL5 regulates various TGN processes through the recruitment of different effectors.

## Results

### Identification of ARMH3 as a potential ARL5 effector

To identify potential effectors of the ARL5A and ARL5B GTPases, we used the MitoID proximity biotinylation method previously employed to identify interactors of RAB[11] and ARL8A/ARL8B GTPases[12]. MitoID involves targeting a bait protein to mitochondria, leading to the recruitment and biotinylation of potential interactors. This technique enriches cytosolic and peripheral-membrane protein interactors but is less suitable for the identification of transmembrane protein interactors. Bait constructs were made by fusing sequences encoding a mitochondrial-targeting sequence (MTS) from the outer mitochondrial membrane protein TOM20, along with the biotin ligase BioID2, to the N-terminus of constitutively active (Q70L) or inactive (T30N) forms of human ARL5A or ARL5B lacking the N-terminal amphipathic α-helix that integrates into membranes (Mito-ARL5 constructs) (Fig. 1a). MTS-BioID2 was used as a negative control (Fig. 1a). Upon transient expression of these constructs in HEK293T cells, the MTS affixed them to mitochondria, while BioID2 catalyzed the biotinylation of neighboring proteins (Fig. 1b). Cell extracts were incubated with Neutravidin-agarose beads and bound proteins were identified by mass spectrometry. Data from three biological replicates per sample were analyzed by plotting the $-\log_{10}$(p-value) *vs.* $\log_2$(fold change) of

proteins identified in the Mito-ARL5 datasets relative to the control dataset (Fig. 1c) (see also Supplementary Data 1). Because effectors preferentially bind to the active forms of GTPases, proteins with a greater relative abundance in the ARL5A and ARL5B Q70L datasets relative to T30N datasets were considered for further analyses. Notable among these proteins was the VPS52 subunit of the GARP complex (Fig. 1c), consistent with its previous identification as an ARL5 effector[6,7]. This finding validated MitoID as an unbiased, discovery-based approach for the identification of ARL5 effectors.

By far the most prominent hit in both ARL5A and ARL5B Q70L, but not ARL5A and ARL5B T30N, datasets was a 689-amino-acid protein known as C10orf76, also termed Armadillo-like helical domain-containing protein 3 (ARMH3, the name used here) (Fig. 1c). Despite its prevalence in our datasets, this protein had not been identified in previous screens for ARL5 interactors[6,13,22,23]. ARMH3 is a highly conserved protein thought to have been present in the last eukaryotic common ancestor (LECA)[24], and is ubiquitously expressed across human tissues and cells (proteinatlas.org)[25,26]. Previous studies showed that ARMH3 localizes to the Golgi complex, where it regulates the activity of Golgi PI4KB[14,27,28]. The possibility that ARMH3 could link ARL5 to PI4KB regulation prompted us to characterize the ARL5-ARMH3 interaction in more detail.

### Characterization of the interaction of ARMH3 with ARL5

To validate a potential interaction between ARMH3 and active ARL5, we examined the redistribution of GFP-tagged ARMH3 (GFP-ARMH3) upon co-expression with different Mito-ARL5 constructs in HeLa cells (Fig. 2a). Immunofluorescence microscopy showed that GFP-ARMH3 localized to the Golgi region in cells expressing inactive Mito-ARL5A-T30N or Mito-ARL5B-T30N, but redistributed to mitochondria in cells expressing constitutively active Mito-ARL5A-Q70L or Mito-ARL5B-Q70L (Fig. 2a, b). GFP, used as a negative control, remained cytosolic in cells expressing the Q70L constructs (Fig. 2a, b).

The structure of ARMH3 has not been experimentally determined. However, secondary structure and AlphaFold2 analyses[29,30] predict a predominantly α-helical structure, with a long loop near the middle of its sequence (Fig. S1a, b). The connection between the N-terminal and C-terminal halves also involves a C-terminal segment that folds over the N-terminal half of the protein (Fig. S1b, arrow). Therefore, ARMH3 is structured as a single fold that cannot be split by cleavage of the middle loop. Consistent with this prediction, GFP-tagged constructs corresponding to the N- and C-terminal halves of ARMH3 were cytosolic (Fig. S1c) and were not redistributed to mitochondria by active Mito-ARL5A-Q70L or Mito-ARL5B-Q70L (Fig. S1d).

To further corroborate the ARL5-ARMH3 interactions, we performed yeast-two hybrid (Y2H) assays by co-expressing different forms of ARL5 fused to the Gal4 DNA-binding domain (BD), and ARMH3 fused to the Gal4 transcriptional activation domain (AD). The ARL5 constructs used in these assays were designed without the N-terminal α-helix (Δ13) to prevent their interaction with membranes and thus enable their import into the nucleus for Gal4 transactivation. Importantly, the yeast *Saccharomyces cerevisiae* does not express ARL5 or ARMH3 orthologs, thereby precluding interference by the corresponding endogenous proteins. In these assays, yeast growth in the absence of histidine and the presence of 3-amino-1,2,4-triazole (3-AT) is indicative of interactions. Using this methodology, we found that ARMH3 interacted with the wild-type (WT) and Q70L, but not T30N, forms of both ARL5A and ARL5B (Fig. 2c, d). In contrast, ARMH3 did not interact with the active forms of two other TGN-localized ARL GTPases, ARL1-Q71L and ARFRP1-Q79L (Fig. 2c). As additional controls, we showed that the SV40 large T antigen (T-Ag) did not interact with any of the above constructs but interacted with its cognate transcription factor p53 (Fig. 2c).

Taken together, the MitoID, mitochondrial relocalization, and Y2H results demonstrated that ARMH3 specifically interacts with the

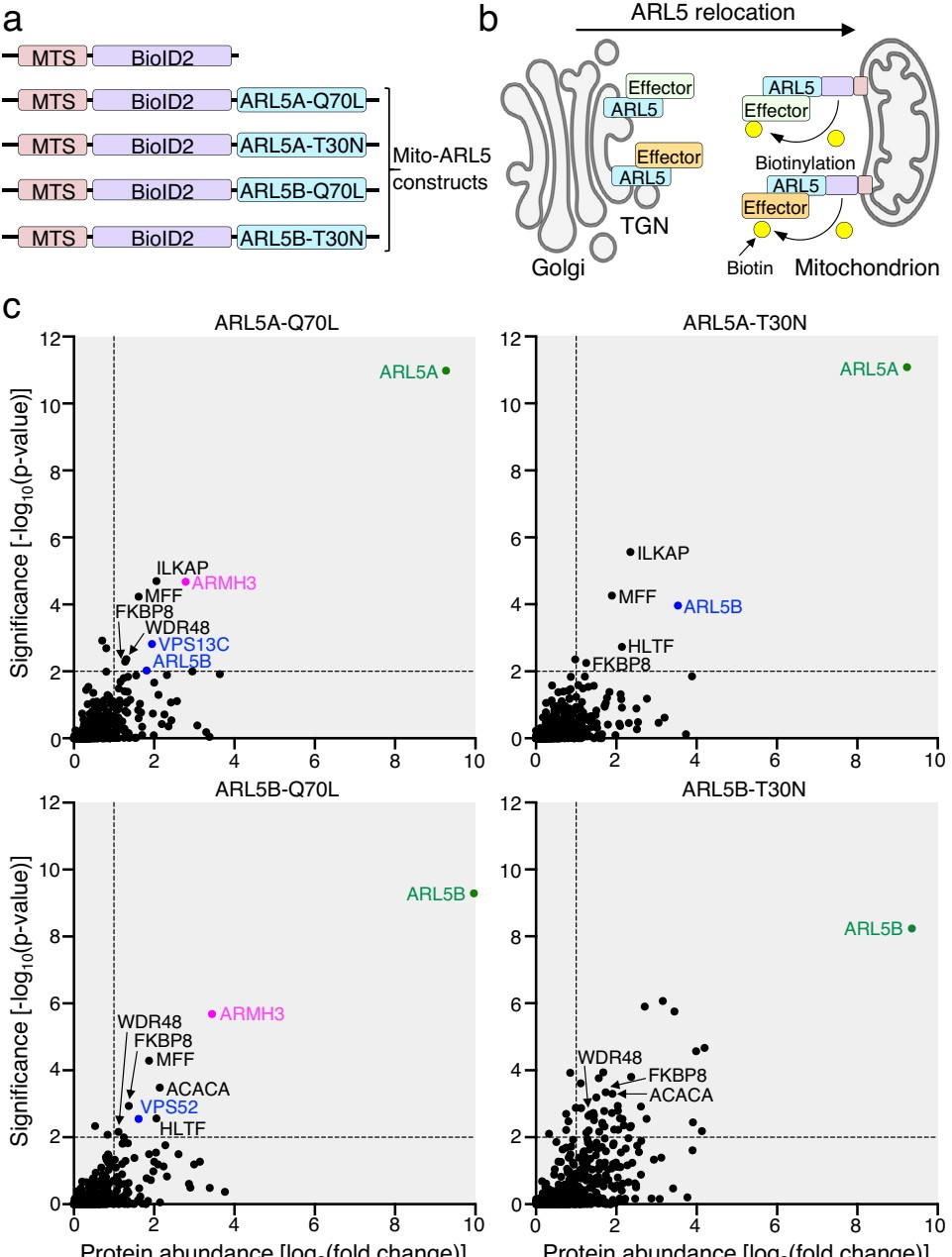

**Fig. 1 | MitoID identifies ARMH3 as a potential ARL5 effector. a** Schematic representation in *N*- to *C*-terminal direction of control and Mito-ARL5 constructs used in MitoID. Constructs comprise the mitochondrial targeting sequence (MTS) from TOM20, followed by the biotin ligase BioID2, with or without constitutively active (Q70L) or inactive (T30N) ARL5A or ARL5B lacking the 13-amino-acid N-terminal α-helix. These constructs were expressed by transient transfection in HEK293T cells. **b** Schematic representation of the MitoID procedure. The Mito-ARL5 or control constructs are targeted to mitochondria, and neighboring proteins biotinylated, affinity-purified, and identified by mass spectrometry. Schematic partially created in BioRender. Bonifacino, J. (2024) BioRender.com/w68r614. **c** Graphs showing the log$_2$(fold-change) of proteins identified in Mito-ARL5 relative to control datasets *vs* the -log$_{10}$(*p*-value). Select hits are labeled in color. ARMH3, VPS52, and VPS13C were identified in the Q70L but not T30N forms of ARL5A and/ or ARL5B, and were thus deemed of interest. In contrast, ACACA, FKBP8, HLTF, ILKAP, MFF, and WDR48 were identified in both the Q70L and T30N forms between ARL5 variants and were not further investigated. Source data are provided in the Source Data file.

active forms of ARL5A and ARL5B, consistent with ARMH3 being a bona fide ARL5 effector.

### Localization of ARMH3 to the Golgi complex is dependent on ARL5, ARFRP1, and SYS1

Next, we investigated if the localization of ARMH3 to the Golgi complex is dependent on ARL5. To this end, we used a previously described[7] CRISPR-Cas9-generated double knockout (KO) line for both ARL5A and ARL5B (referred to as ARL5 KO for simplicity) in

HeLa cells, and determined the localization of GFP-ARMH3 expressed by transient transfection into these cells. We observed that ARL5 KO abrogated the localization of GFP-ARMH3 to the Golgi complex (Fig. 3a, b).

The association of ARL5 with the Golgi complex depends on its upstream activators ARFRP1 and SYS1[7]. Consistent with this dependence, the association of GFP-ARMH3 with the Golgi complex was also abolished in ARFRP1-KO and SYS1-KO HeLa cells (Fig. 3a, b). In contrast, KO of the VPS54 subunit of GARP (another ARL5 effector)[6,7] or ARL1

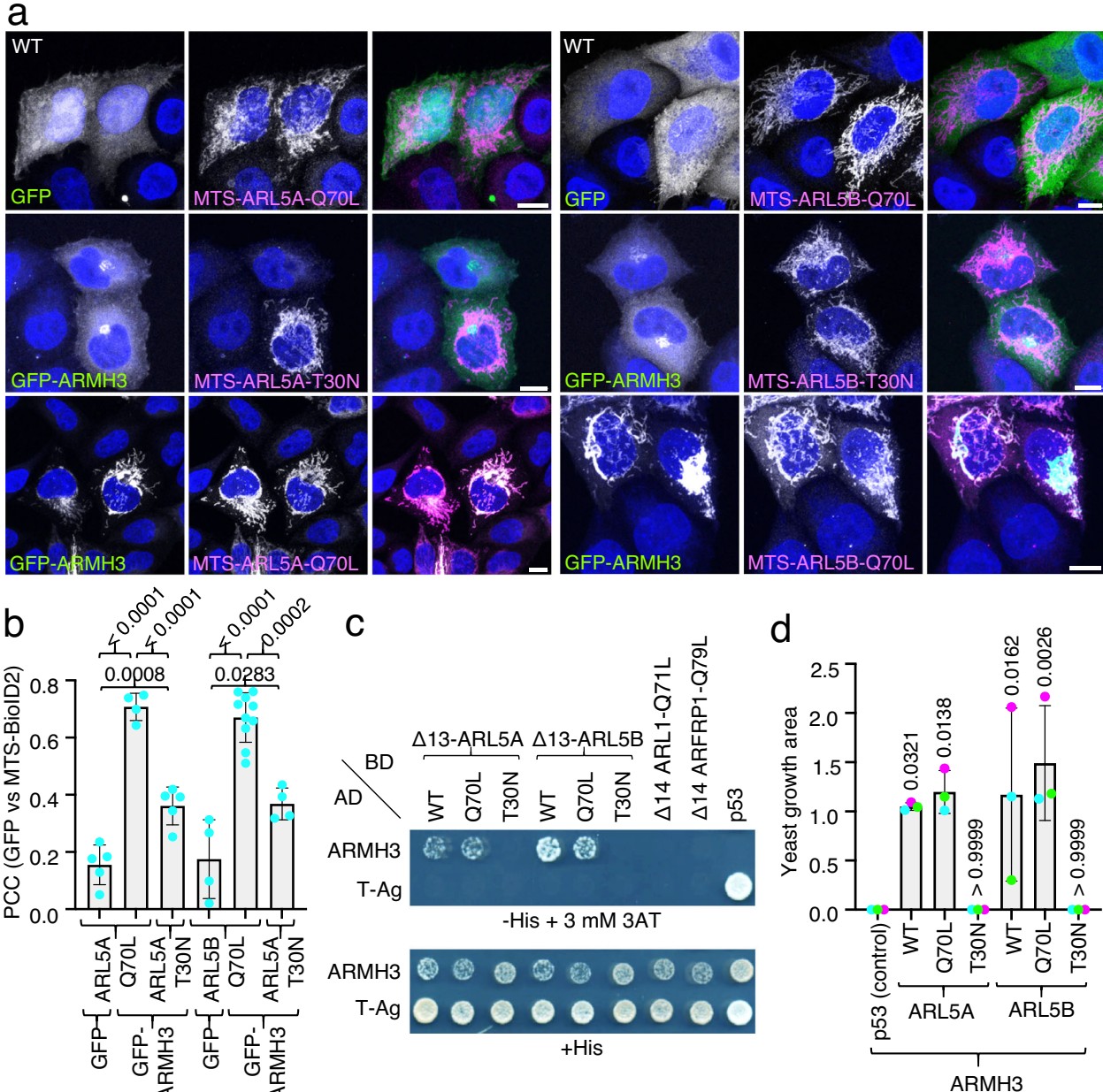

**Fig. 2 | Demonstration of ARL5-ARMH3 interactions using mitochondria-relocalization and yeast-two hybrid assays. a** Mitochondria-relocalization assays. WT HeLa cells were transiently co-transfected with constructs encoding GFP (control) or GFP-ARMH3 (green) and the indicated Mito-ARL5 constructs (magenta). Cells were then fixed, permeabilized, immunolabeled for BioID2, and imaged by confocal fluorescence microscopy. Single-channel images are shown in grayscale with nuclei (DAPI) in blue. Scale bars: 10 μm. Notice the relocalization of GFP-ARMH3 to mitochondria by the Mito-ARL5-Q70L but not -T30N constructs. **b** Pearson's correlation coefficient (PCC) for the co-localization of GFP-ARMH3 or GFP with the indicated Mito-ARL5 constructs calculated from an experiment such as that shown in panel **a**. The graph represents the mean ± SD of values from the following number of cells in the experiment (from left to right): $n$ = 5, 4, 5, 4, 10 and 4. The statistical significance of the differences was calculated by one-way ANOVA with multiple comparisons using Tukey's test. $P$-values are indicated on the graphs.

**c** Y2H analysis of the interaction of the constructs indicated on top fused to the GAL4 DNA-binding domain (BD) and the constructs indicated on the left fused to the GAL4 transcriptional activation domain (AD). The p53 and T-Ag fusion proteins were used as controls. Growth in the absence of histidine (−His) and presence of 3 mM 3-amino-1,2,4 triazole (3AT) is indicative of interactions. Growth in the presence of histidine (+His) is a control. Notice the interaction of ARMH3 with the WT and Q70L forms of ARL5A and ARL5B, and the failure of ARMH3 to interact with the active forms of ARL1 and ARFRP1. **d** Quantification of the colony area from the Y2H assays shown in panel **c**. The graph represents the mean ± SD from three independent, color-coded experiments. The statistical significance of the differences was calculated by one-way ANOVA with multiple comparisons to the interaction between p53 and ARMH3 (control) using Dunnett's test. $P$-values are indicated on the graphs. Source data are provided in the Source Data file.

(another ARF-family GTPase activated by ARFRP1 and SYS1)[7,31–33] had no effect on the localization of GFP-ARMH3 to the Golgi complex (Fig. 3a, b).

The localization of GFP-ARMH3 to the Golgi complex in ARL5-KO HeLa cells could be rescued by co-expression of WT or Q70L, but not

T30N, ARL5B-mCherry (Fig. 3c, d). In contrast, GFP-ARMH3 remained cytosolic upon expression of the analogous WT, Q79L, or T31N forms of ARFRP1 in ARL5-KO cells (Fig. 3d).

Consistent with the notion that ARMH3 is an ARL5 effector, we also observed competition in the recruitment of another ARL5

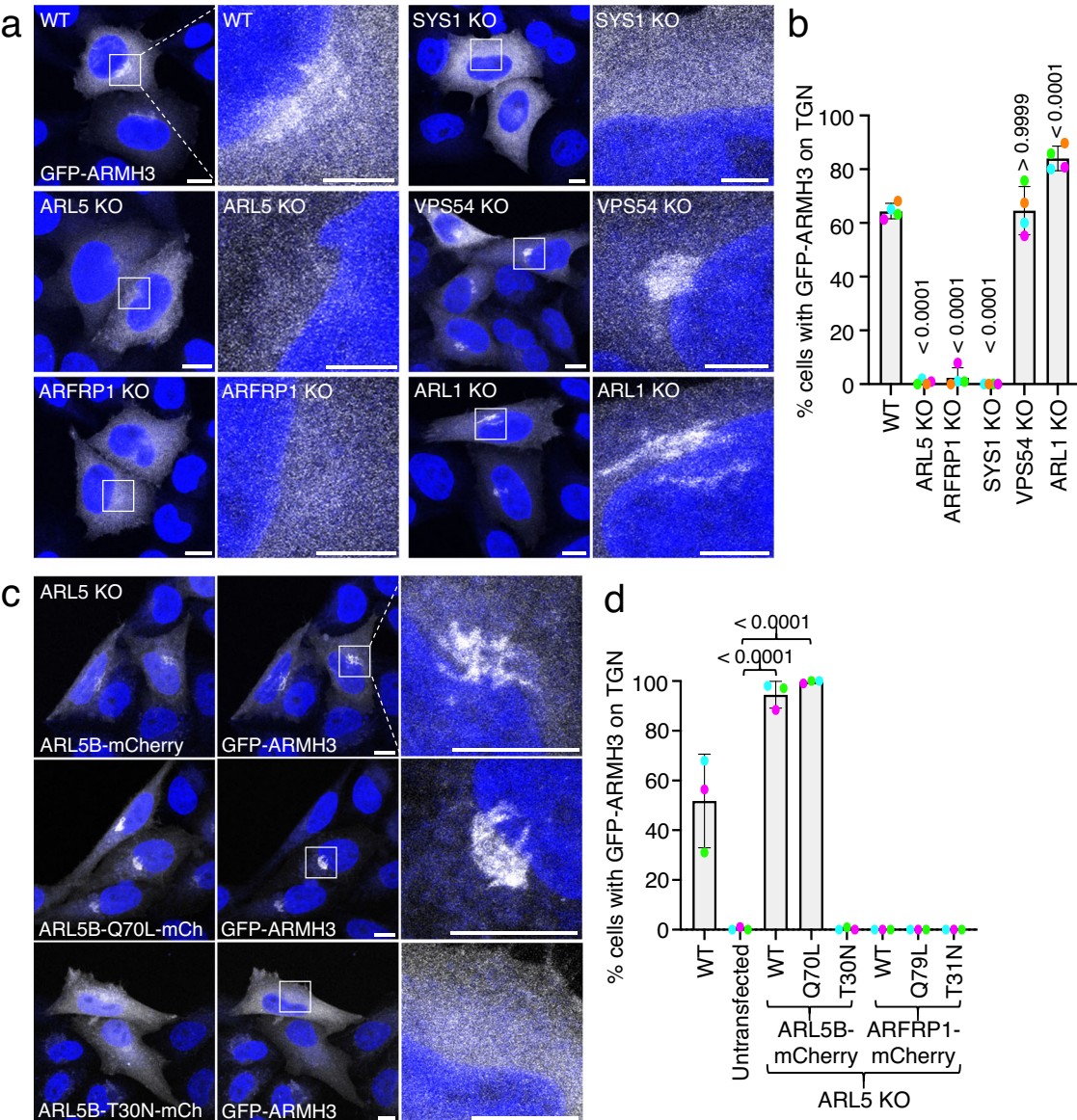

**Fig. 3 | Association of ARMH3 with the Golgi complex is dependent on ARL5 and its upstream activators ARFRP1 and SYS1. a** Confocal fluorescence microscopy of GFP-ARMH3 expressed by transient transfection in the indicated WT and KO HeLa cell lines. GFP-ARMH3 is shown in grayscale and nuclei (DAPI) in blue. Each pair of images shows a lower magnification of a larger field and a higher magnification of the boxed area. Scale bars: 10 μm. Notice the dissociation of GFP-ARMH3 from the Golgi complex in cells with KO of ARL5, ARFRP1, or SYS1. **b** Quantification of the percentage of cells with GFP-ARMH3 at the Golgi complex from experiments such as that shown in panel a. Values are the mean ± SD from four independent, color-coded experiments. The statistical significance of the differences was calculated by one-way ANOVA with multiple comparisons using Dunnett's test. *P*-values are indicated on the graphs. **c** Rescue of GFP-ARMH3 Golgi localization. ARL5-KO HeLa cells were co-transfected with plasmids encoding WT, Q70L, or T30N forms of ARL5B-mCherry, and GFP-ARMH3. Cells were imaged by confocal fluorescence microscopy. Images are shown in grayscale with nuclei (DAPI) in blue. Higher magnifications of the boxed areas are shown on the right column. Scale bars: 10 μm. Notice the rescue of GFP-ARMH3 Golgi localization by WT and Q70L, but not T30N, ARL5B-mCherry. **d** Quantification of the percentage of cells with GFP-ARMH3 at the Golgi complex from experiments such as that in panel **b**. *P*-values are indicated on the graphs. Source data are provided in the Source Data file.

effector, VPS54-13myc[7], to the Golgi complex upon expression of GFP-ARMH3 (Fig. S2a, b).

Based on these results, we concluded that ARMH3 requires active ARL5 and its upstream activators ARFRP1 and SYS1 for its association with the Golgi complex.

**ARMH3 enhances the interaction of ARL5 with PI4KB**

To investigate the ARL5-dependent function of ARMH3 at the Golgi complex, we sought to identify ARMH3 interactors using MitoID with MTS-BioID2-ARMH3 and MTS-BioID2 (control) constructs (Fig. 4a), as described above for ARL5 (Fig. 1a, b) (see Supplementary Data 2).

Among the hits in the ARMH3 screen was ARL5B, albeit with relatively low abundance (Fig. 4b). Other hits included proteins that had been previously identified as ARMH3 interactors, such as the Golgi-specific Brefeldin-A resistance factor 1 (GBF1, an ARF1 GEF)[24] and PI4KB[14,27,28,34,35] (Fig. 4b, c). The significance of other hits in this screen is unknown. Because of the potential for ARMH3 to link ARL5 with PI4KB, we focused our next experiments on PI4KB.

In Y2H assays, ARMH3 exhibited strong interactions with the WT and Q70L, but not T30N, forms of ARL5A and ARL5B (Fig. 4d), as already shown in Fig. 2c, d. In contrast, both the N-terminal domain of PI4KB (PI4KB-N) and full-length PI4KB showed weak or no interaction

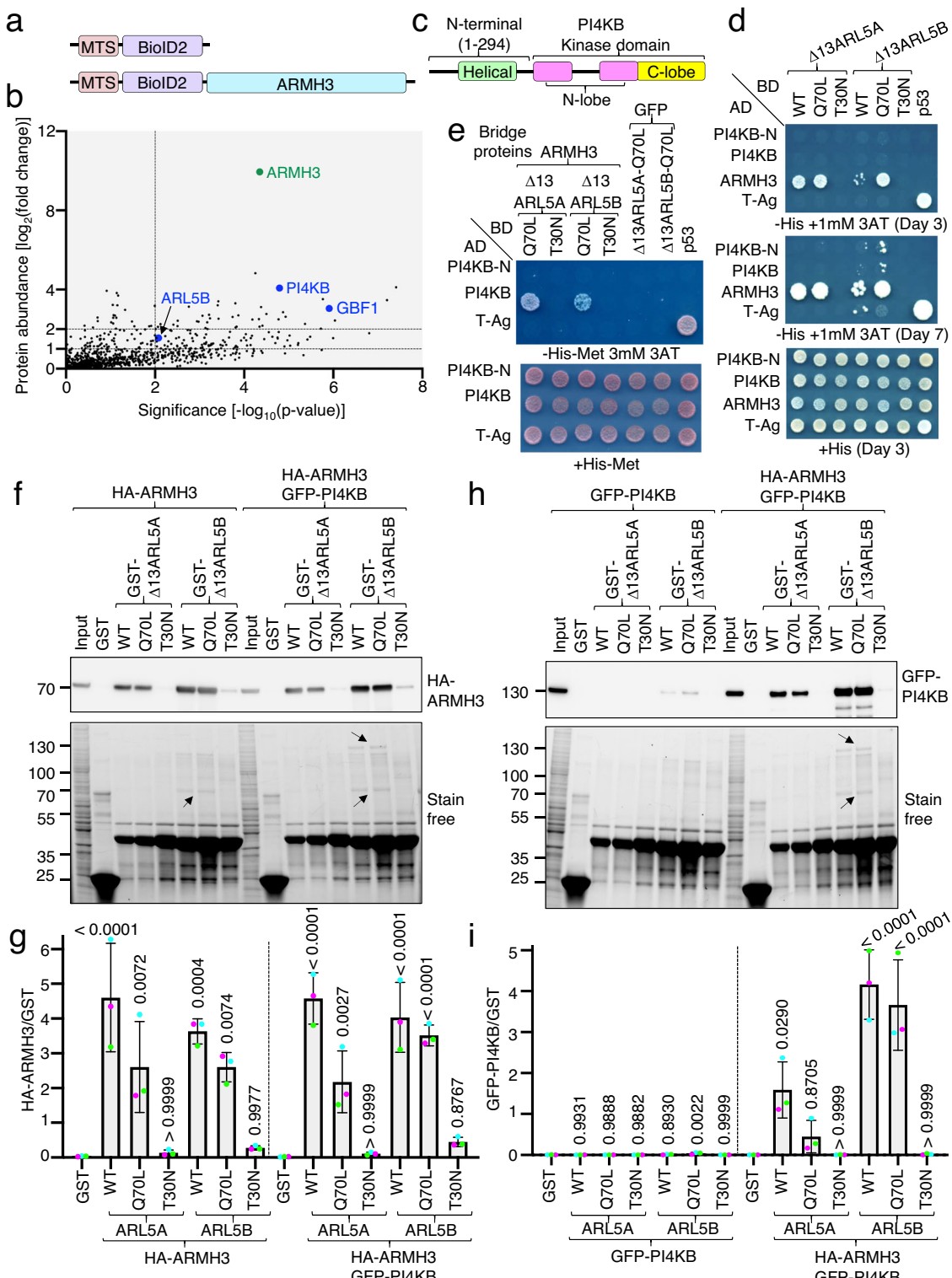

with all ARL5 forms (Fig. 4d). To examine whether ARMH3 could enable the interaction of PI4KB with ARL5, we used a yeast three-hybrid (Y3H) system involving co-transformation with (i) a pBridge plasmid encoding different forms or ARL5 fused to the Gal4 DNA-binding domain (BD) in multiple cloning site I, and ARMH3 or GFP (control) in multiple cloning site II, and (ii) a pGADT7 plasmid encoding PI4KB-N or full-length PI4KB fused to the Gal4 transcription activation domain (AD) (Fig. 4e). These assays showed that full-length PI4KB, but not PI4KB-N, interacted with the Q70L, but not T30N, forms of ARL5A and ARL5B, but only in the presence of ARMH3 (Fig. 4e).

To confirm these findings with an alternative assay, we conducted GST pulldowns using GST fused to various forms of ARL5, and extracts from HeLa cells stably expressing HA-ARMH3 and GFP-PI4KB, either individually or in combination (Fig. 4f–i). We observed that the WT and Q70L, but not T30N, forms of ARL5A and ARL5B pulled down large amounts of HA-ARMH3 (Fig. 4f, g), but little or no GFP-PI4KB (Fig. 4h, i) when the latter two were expressed individually. However, when HA-ARMH3 and GFP-PI4KB were co-expressed, both proteins came down strongly with the WT and Q70L, but not T30N, forms of ARL5A and ARL5B (Fig. 4f–i).

**Fig. 4 | ARMH3 enhances the interaction of ARL5 with PI4KB. a** Schematic representation of control and MTS-BioID2-ARMH3 constructs. **b** Plot of the -log₁₀(*p*-value) vs the log₂(fold-change) of proteins identified by mass spectrometry for MTS-BioID2-ARMH3 relative to control. Notice the presence of colored ARL5B, PI4KB, and GBF1 among the interacting proteins. **c** Schematic representation of PI4KB. **d** Y2H of (i) Δ13ARL5 variants fused to the GAL4 DNA-binding domain (BD) with (ii) ARMH3 or PI4KB variants fused to the GAL4 transcriptional activation domain (AD). Notice the interaction of WT and Q70L, but not T30N, ARL5A/B with ARMH3 but not full-length or N-terminal domain of PI4KB (PI4KB-N). **e** Y3H of (i) Δ13ARL5 variants fused to the GAL4 DNA-binding domain (BD), with (ii) PI4KB variants fused to the GAL4 transcriptional activation domain (AD), in the presence of (iii) ARMH3 or GFP (negative control). Notice the interaction of ARL5A/B Q70L with full-length PI4KB, but only in the presence of ARMH3. **f** Pulldowns using GST-Δ13ARL5 variants and detergent extracts from HeLa cells stably expressing HA-ARMH3, with and without GFP-PI4KB. HA-ARMH3 was detected by immunoblotting

with an antibody to HA, and total proteins with Stain-Free detection. Positions of molecular mass markers (kDa) are indicated on the left. The positions of HA-ARMH3 and GFP-PI4KB on the Stain-Free images are indicated by arrows. Notice the pulldown of HA-ARMH3 or the HA-ARMH3–GFP-PI4KB complex by WT and Q70L forms of both ARL5A and ARL5B. **g** Quantification of HA-ARMH3 pulled down in panel **f**. Values are the mean ± SD from three independent, color-coded experiments. Significance was calculated by one-way ANOVA with multiple comparisons using Dunnett's test. *P*-values are indicated on the graphs. **h** Pulldowns using GST-Δ13ARL5 variants and detergent extracts from HeLa cells stably expressing GFP-PI4KB with or without HA-ARMH3. Pulldowns were performed as described for panel f, but with GFP-PI4KB detected with an antibody to GFP. Notice the pulldown of GFP-PI4KB by WT and Q70L forms of both ARL5A and ARL5B, only when HA-ARMH3 was co-expressed. **i** Quantification of GFP-PI4KB pulled down in panel h, as described for panel g. Source data are provided in the Source Data file.

We also examined the effect of ARMH3 on the ARL5-PI4KB interaction using microscopy-based assays. A mitochondria-relocalization assay using WT HeLa cells showed Mito-ARL5B-Q70L re-localized GFP-PI4KB from the Golgi complex to mitochondria, only in the presence of HA-ARMH3 (Fig. 5a, b). In contrast, a GFP-ARMH3-FLH409AAA mutant that fails to interact with PI4KB[27] did not lead to mCherry-PI4KB re-localization to mitochondria by Mito-ARL5B-Q70L (Fig. S3a, b). Moreover, expression of ARL5B-Q70L-mCherry in WT HeLa cells increased the association of endogenous PI4KB with the Golgi complex, an effect that was further enhanced upon co-expression of HA-ARMH3 (Fig. 5c, d). Immunofluorescent staining and structured illumination microscopy (SIM) of HeLa cells stably expressing ARL5B-Q70L-mCherry, HA-ARMH3, and GFP-PI4KB revealed a high degree of co-localization of all three proteins at the Golgi complex (Fig. S4a, b), consistent with a tripartite interaction.

Taken together, these assays demonstrated that ARMH3 enables the interaction of PI4KB with ARL5, and enhances the recruitment of PI4KB to the Golgi complex.

## ARL5 and ARMH3 are required for PI4KB activity at the Golgi complex

Our next experiments aimed to investigate whether ARL5 and ARMH3 are required for PI4KB recruitment and subsequent PI4P generation at the Golgi complex. Immunofluorescence microscopy with an antibody to endogenous PI4KB showed that PI4KB KO in HeLa cells (Fig. 6a) abolished staining at the Golgi complex, indicating that this staining was specific (Fig. 6b). Furthermore, we observed that PI4KB localization to the Golgi complex was slightly decreased in ARL5-KO cells but was not significantly altered in ARMH3 KO (Fig. 6b, c). These findings are consistent with previous studies that demonstrated either partially decreased or unchanged staining for PI4KB staining in ARL5-KO or ARMH3-KO cells[13,14,27]. Recruitment of GFP-ARMH3 to the Golgi complex was also not affected in PI4KB-KO cells (Fig. S5a).

Overexpression of ARL5B-Q70L-GFP in ARMH3-KO cells did not increase the association of PI4KB with the Golgi complex (Fig. 6d, e), suggesting that the increases observed upon overexpression of ARL5B-Q70L-GFP in WT cells (Fig. 5c, d) are dependent on endogenous ARMH3. Accordingly, co-expression of ARL5B-Q70L-GFP and HA-ARMH3 in ARMH3-KO cells did increase the amount of PI4KB at the Golgi complex (Fig. 6d, e), as also seen in WT cells (Fig. 5c, d). Individual expression of HA-ARMH3 (Fig. 6d, e) or GFP-ARMH3-FLH409AAA (Fig. S5b, c) did not increase Golgi PI4KB staining, indicating that the increase requires the co-expression of active ARL5B and WT ARMH3. We interpret these results to mean that there is a basal pool of PI4KB at the Golgi complex that does not require ARL5 or ARMH3 for localization. In the absence of ARL5 or ARMH3, this pool may alternatively associate with the Golgi complex via other PI4KB interactors such as ACBD3[14,36,37].

If ARL5 and ARMH3 are not strictly required for the recruitment of PI4KB to the Golgi complex, what might then be the function of the ARL5-ARMH3-PI4KB axis? PI4KB is the main enzyme responsible for PI4P synthesis at the Golgi complex[38,39]. Immunofluorescence microscopy using an antibody to PI4P[40] revealed a tight cluster of PI4P puncta at the Golgi complex, along with scattered puncta throughout the cytoplasm (Fig. 7a). PI4KB KO significantly reduced staining of PI4P at the Golgi relative to peripheral puncta (Fig. S6a, b), in agreement with previous findings[38,39]. The peripheral puncta probably represent endosomal PI4P synthesized by other phosphatidylinositol 4-kinases, such as PI4K2A and PI4K2B[41,42]. Remarkably, knocking out ARL5, its upstream activators ARFRP1 or SYS1, or ARMH3 also led to a drastic reduction in PI4P staining at the Golgi complex (Fig. 7a–c). Co-staining for the Golgi protein giantin showed that none of these KOs altered the overall morphology of the Golgi complex, demonstrating that the loss of PI4P staining was not due to disruption of Golgi structure (Fig. 7a). Fluorescence microscopy using cells co-expressing the PI4P biosensor mCherry-P4M[39] and the Golgi marker YFP-GalT[43] confirmed the reduction of PI4P levels without alteration of Golgi appearance in ARMH3-KO, ARL5-KO or PI4KB-KO cells (Fig. S6c).

From these experiments, we concluded that ARL5 and ARMH3, along with their upstream activators ARFRP1 and SYS1, are critical for PI4KB activity at the Golgi complex.

## ARMH3 and PI4KB are not required for retrograde transport from endosomes to the TGN

ARL5 and its effector GARP were previously shown to participate in the retrograde transport of various cargos from endosomes to the TGN in mammalian cells[6,7,44–47]. To explore the potential involvement of the ARL5-ARMH3-PI4KB axis in retrograde transport, we analyzed the intracellular distribution of the retrograde cargo proteins TGN46 and CI-MPR in HeLa cells with KO of ARL5 or its effectors. We observed that ARL5-KO and VPS54-KO HeLa cells exhibited reduced TGN46 localization to the TGN (Fig. S7a) and total cellular levels (Fig. S7b), consistent with the previously reported dispersal and degradation of this protein when it cannot recycle to the TGN[7,44,47–49]. However, there was more CI-MPR at the TGN in the absence of ARL5 or VPS54 (Fig. S7a). This phenomenon can be attributed to the tethering of endosome-derived retrograde carriers to the TGN by ARL1 and its effector golgins (GCC88, Golgin-245, and Golgin-97) when the carriers cannot undergo SNARE-dependent fusion[7]. In contrast to KO of ARL5 or VPS54, KO of ARMH3 or PI4KB had no effect on TGN46 or CI-MPR distribution and levels (Fig. S7a, b).

STxB[50] is another retrograde cargo that undergoes internalization from the plasma membrane, followed by ARL5- and GARP-dependent delivery to the TGN[7,44]. We observed that ARL5 KO or VPS54 KO decreased the amount of internalized Cy3-conjugated STxB (Cy3-STxB) delivered to the TGN (Fig. S7c), as previously reported[7,44]. In

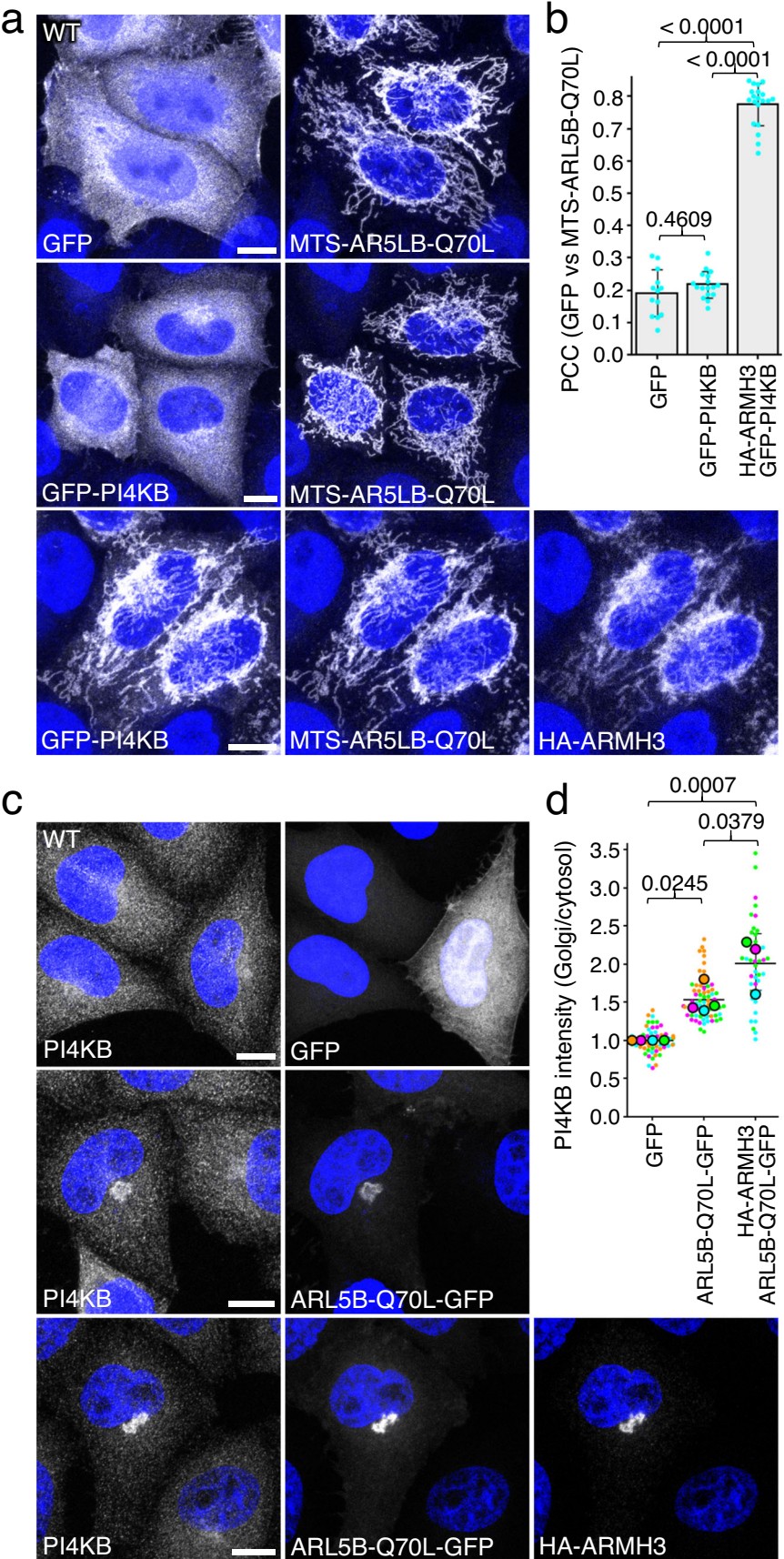

**Fig. 5 | ARMH3 enhances the recruitment of PI4KB by ARL5. a** Mitochondria-relocalization assay. WT HeLa cells were co-transfected with plasmids encoding GFP or GFP-PI4KB in combination with Mito-ARL5B-Q70L, minus or plus HA-ARMH3, as indicated in the figure. Cells were then fixed, permeabilized, immunostained for GFP, BioID2, and HA, and imaged by confocal microscopy. Images are shown in grayscale with nuclei (DAPI) in blue. Scale bars: 10 μm. Notice the recruitment of GFP-PI4KB to mitochondria upon co-expression of HA-ARMH3. **b** Quantification of the Pearson's correlation coefficient (PCC) of GFP *vs* BioID2 from the experiment shown in panel a. The graph represents the mean ± SD of values from the following number of cells in the experiment (from left to right): $n = 13$, 17, and 20, as indicated in the figure. The statistical significance of the differences was calculated by one-way ANOVA with multiple comparisons using Tukey's test. *P*-values are indicated on the graphs. **c** Golgi-recruitment assay. WT

HeLa cells were co-transfected with plasmids encoding GFP or ARL5B-Q70L-GFP, minus or plus HA-ARMH3, as indicated in the figure. Cells were then fixed, permeabilized, immunostained for endogenous PI4KB, and imaged by confocal microscopy. Images are shown in grayscale with nuclei (DAPI) in blue. Scale bars: 10 μm. Notice the progressive increase in PI4KB association with the Golgi complex upon expression of ARL5B-Q70L-GFP, and ARL5B-Q70L-GFP plus HA-ARMH3. **d** Quantification of PI4KB intensity from the following number of the experiments (from left to right): $n = 4$, 4, and 3, such as that shown in panel c. Data are represented as SuperPlots showing the mean ± SD of the means of the individual data points. The statistical significance of the differences was calculated by one-way ANOVA with multiple comparisons using Tukey's test. *P*-values are indicated on the graphs. Source data are provided in the Source Data file.

contrast, ARMH3-KO or PI4KB-KO cells exhibited unchanged or even elevated levels of Cy3-STxB at the TGN (Fig. S7c).

These experiments thus demonstrated that ARMH3 and PI4KB are dispensable for the retrograde transport of various ARL5- and GARP-dependent cargos to the TGN.

## ARMH3 and PI4KB regulate GOLPH3 recruitment and glycan modifications at the TGN

What processes are then regulated by ARMH3 and PI4KB at the Golgi complex? Because these proteins mediate PI4P synthesis at the Golgi complex, we hypothesized that they promote the association of PI4P-binding proteins with the TGN. One group of PI4P-binding, TGN-associated proteins are the coat proteins AP-1[51–53] and GGA3[54]. However, we observed that KO of ARL5, ARMH3, or PI4KB did not alter their localization at the TGN (Fig. S8). This negative result is likely due to both AP-1[53,55] and GGA3[56,57] binding to active ARF1 and their association with PI4K2A, not PI4KB[51,54]. Furthermore, we found that KO of ARL5, ARMH3, or PI4KB did not alter the Golgi localization of the coat protein COPI and the PI4KB-recruitment factor ACBD3 (Fig. S8), neither of which is known to bind PI4P. In contrast, these KOs did decrease the association of another PI4P-binding protein, GOLPH3, with the TGN (Fig. 8a, b), without causing changes in the total levels of GOLPH3 (Fig. S7b). The staining of GOLPH3 at the Golgi complex in ARL5-KO, ARMH3-KO, or PI4KB-KO cells could be rescued by re-expression of the corresponding GFP-tagged proteins (Fig. S9).

GOLPH3 (originally named GPP34 or GMx33)[58,59] and its yeast ortholog Vps74p[60] have been shown to perform various functions, the best-defined of which is serving as a COPI adaptor for the retention and stabilization of glycosylation enzymes at the Golgi complex[19,21,61–63]. Accordingly, depletion of GOLPH3 or its paralog GOLPH3L causes changes in the glycosylation of both Golgi-resident and itinerant proteins[21]. Moreover, misregulation of GOLPH3-mediated post-Golgi trafficking and glycosylation have significant consequences in promoting tumorigenesis[15,64].

We therefore used a lectin-blot assay[47,65] to examine the global effect of knocking out ARL5 or its effectors on carbohydrate modifications of total cellular proteins. We observed that KO of ARL5 or VPS54 caused a large increase in overall signal using the lectins *Helix pomatia* agglutinin (HPA) and peanut agglutinin (PNA), which recognize terminal *N*-acetylgalactosamine and D-galactose-β-(1 → 3) *N*-acetyl-D-galactosamine on O-glycan chains, respectively (Fig. 8c). This observation was consistent with the previously demonstrated requirement of the GARP complex for the retention and stabilization of glycosylation enzymes at the Golgi complex, and for the resulting carbohydrate modifications of various glycoproteins[7,47,49]. KO of ARMH3 or PI4KB caused smaller but reproducible increases in the overall intensity of glycoproteins on the HPA and PNA blots (Fig. 8c, red arrows).

Finally, we also found that KO of ARL5, ARMH3, or PI4KB decreased the electrophoretic mobility of LAMP1 relative to WT cells (Fig. 8d, e). These changes were reversed by stable rescue of ARL5,

ARMH3, or PI4KB in the corresponding KO lines (Fig. 8d). Treatment with PNGase F, which cleaves the *N*-glycans of glycoproteins, eliminated these differences, indicating that they were in fact due to differences in *N*-glycosylation (Fig. 8e). KO of VPS54 increased the intensity, although it did not change the electrophoretic mobility of LAMP1, as previously observed in fibroblasts from a patient with mutations in the VPS51 subunit of GARP and the related complex EARP[49].

These findings collectively demonstrate that ARL5, ARMH3, and PI4KB promote glycan processing at the Golgi complex, likely by recruiting the PI4P-binding protein GOLPH3. We note that depleting these proteins has a milder effect compared to GOLPH3/GOLPH3L KO[21] or GARP KO[66], likely because GOLPH3 dissociation from the TGN is only partial in ARL5-, ARMH3-, or PI4KB-KO cells.

## Discussion

Our study identified ARMH3 as an ARL5 effector and demonstrated the existence of a SYS1-ARFRP1-ARL5-ARMH3 axis that is critical for PI4KB-dependent maintenance of PI4P at the Golgi complex (Fig. 8f). This PI4P pool in turn contributes to the association of the PI4P-binding oncoprotein GOLPH3 with the Golgi complex (Fig. 8f). In line with the known role of GOLPH3 in maintaining the Golgi glycosylation machinery[16–21], ARL5, ARMH3, and PI4KB also promote terminal *O*- and *N*-linked glycan modifications on glycoproteins.

Several lines of evidence support the role of ARMH3 as an ARL5 effector. First, ARMH3 preferentially interacts with the active form of ARL5, as demonstrated by MitoID (Fig. 1c), mitochondrial relocalization (Fig. 2a, b), Y2H (Fig. 2c, d), and in vitro pulldown assays (Fig. 4f, g). Furthermore, the localization of ARMH3 to the Golgi complex is dependent on active ARL5 (Fig. 3a–d). Finally, SYS1 and ARFRP1 are also required for the association of ARMH3 with the Golgi complex (Fig. 3a, b), consistent with their role in recruiting and promoting the activation of ARL5 at this location[7].

Previous studies reported interactions of ARL5 with PI4KB[13], and of ARMH3 with PI4KB[14,27,28]. Our finding that ARMH3 is an ARL5 effector integrates these observations within the framework of a tripartite ARL5-ARMH3-PI4KB complex. In this complex, ARMH3 facilitates the interaction of ARL5 with PI4KB (Fig. 4d-i), potentially serving as a bridge or allosteric enhancer. Without ARMH3, the interaction of ARL5 with PI4KB is weak or absent. In this context, the previously reported proximity labeling and co-immunoprecipitation of PI4KB with ARL5[13] could be due to the presence of ARMH3 within the cells. This would explain the observation that ARL5 activates PI4P synthesis at the Golgi complex in cellulo, but not when using isolated components in vitro[13]. Elucidating the precise molecular mechanism underlying the ARMH3-dependent interaction of ARL5 with PI4KB will require structural analysis of the tripartite complex.

PI4KB is predominantly associated with the Golgi complex[67,68]. Intriguingly, our experiments employing KO cell lines revealed that ARL5 and ARMH3 are largely dispensable for the Golgi recruitment of PI4KB (Fig. 6b, c). ARL5 KO caused a partial decrease in Golgi PI4KB,

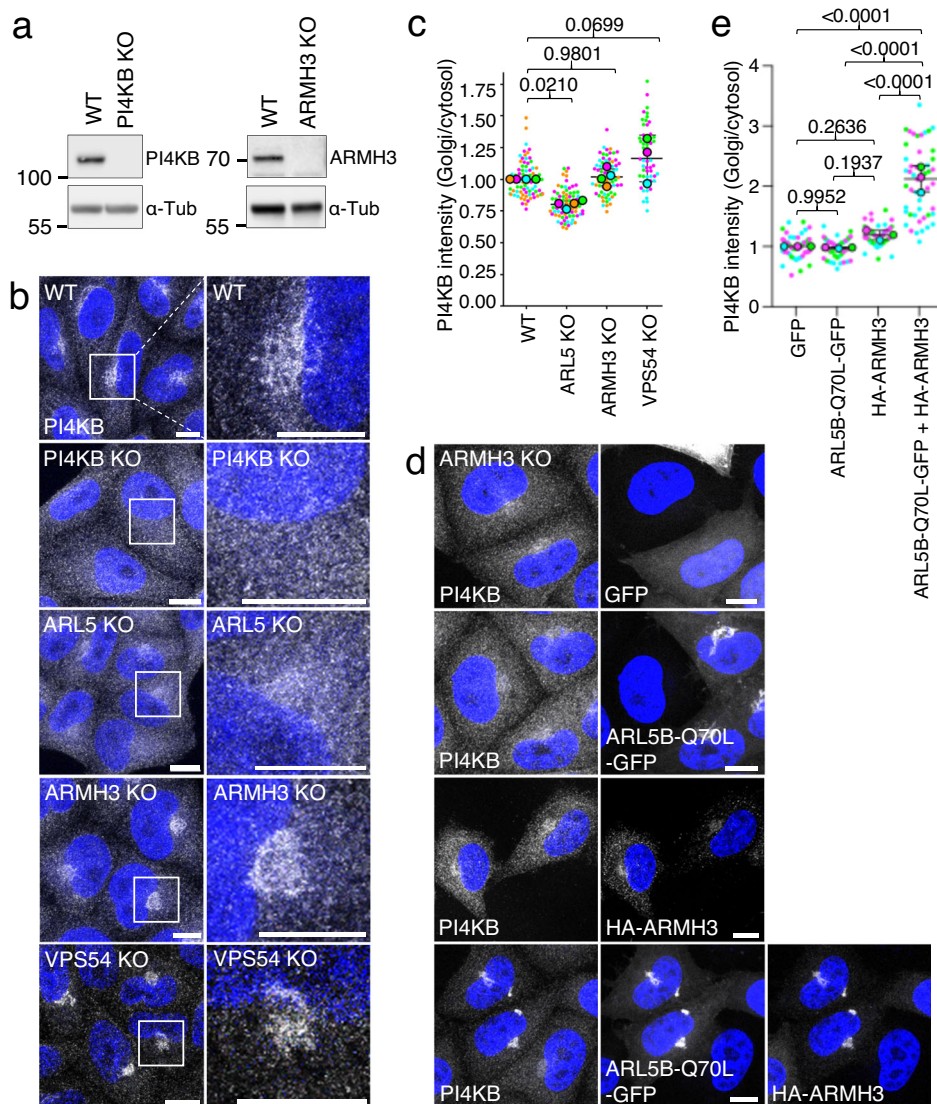

**Fig. 6 | PI4KB associates with the Golgi complex independently of ARL5 and ARMH3. a** Immunoblot analysis of WT, PI4KB-KO, and ARMH3-KO HeLa cells with antibodies to PI4KB and ARMH3, as shown in the figure. The positions of molecular mass markers (kDa) are indicated on the left. The experiments were repeated twice for confirming ARMH3-KO and three times for PI4KB-KO. **b** WT and KO HeLa cells were immunostained for endogenous PI4KB. Images are shown in grayscale with nuclei (DAPI) in blue. The column on the right shows higher magnification images of the boxed areas to their left. Scale bars: 10 μm. Notice the slight decrease in PI4KB staining in ARL5-KO cells and the normal staining of PI4KB in ARMH3-KO cells. **c** Quantification of PI4KB intensity at the Golgi complex relative to the cytosol from experiments such as that shown in panel b. Data are represented as SuperPlots showing the mean ± SD of the means of the individual data points from the following number of the experiments (from left to right): $n = 4, 4, 4,$ and 3. The statistical significance of the differences was calculated by one-way ANOVA

with multiple comparisons using Tukey's test. *P*-values are indicated on the graphs. **d** ARMH3-KO HeLa cells were individually transfected or co-transfected with plasmids encoding GFP, ARL5B-Q70L-GFP, and HA-ARMH3, as indicated in the figure. Cells were then fixed, permeabilized and immunostained for endogenous PI4KB. Images are shown in grayscale with nuclei (DAPI) in blue. Scale bars: 10 μm. Notice the increased PI4KB association with the Golgi complex only when both ARL5B-Q70L-GFP and HA-ARMH3 were co-expressed in ARMH3-KO cells. **e** Quantification of PI4KB intensity at the Golgi complex relative to the cytosol from three experiments such as that shown in panel d. Data are represented as SuperPlots showing the mean ± SD of the means of the individual data points from three independent, color-coded experiments. The statistical significance of the differences was calculated by one-way ANOVA with multiple comparisons using Tukey's test. *P*-values are indicated on the graphs. Source data are provided in the Source Data file.

whereas ARMH3 KO had no significant effect, in agreement with previous observations[14,27,28]. Additionally, we observed that PI4KB KO did not prevent the association of ARMH3 with the Golgi complex (Fig. S5a). The persistence of PI4KB at the Golgi complex in the absence of ARL5 and ARMH3 could be due to other PI4KB-binding proteins such as ACBD3[14,36]. Indeed, a previous study showed that ARMH3 and ACBD3 differentially recruit PI4KB to distal (*i.e., trans* cisternae and TGN) and proximal regions (*i.e., cis* and medial cisternae) of the Golgi complex, respectively[14]. Only when both ARMH3 and ACBD3 were knocked out did the association of PI4KB with the Golgi complex

diminish[14]. Accordingly, PI4KB could shift to earlier Golgi cisternae in our ARL5-KO or ARMH3-KO cells, a distinction that we could not make with the diffraction-limited confocal microscopy used in our study. We also observed that the localization of ACBD3 to the Golgi complex is independent of ARL5 (Fig. S8), consistent with ACBD3 being an alternative anchor for PI4KB at the Golgi complex. Despite ARL5 and ARMH3 being largely dispensable for PI4KB localization to the Golgi complex, overexpression of these proteins increased Golgi PI4KB staining (Figs. 5 and 6), suggesting that these interactions can nevertheless enhance PI4KB recruitment to the Golgi complex. This is

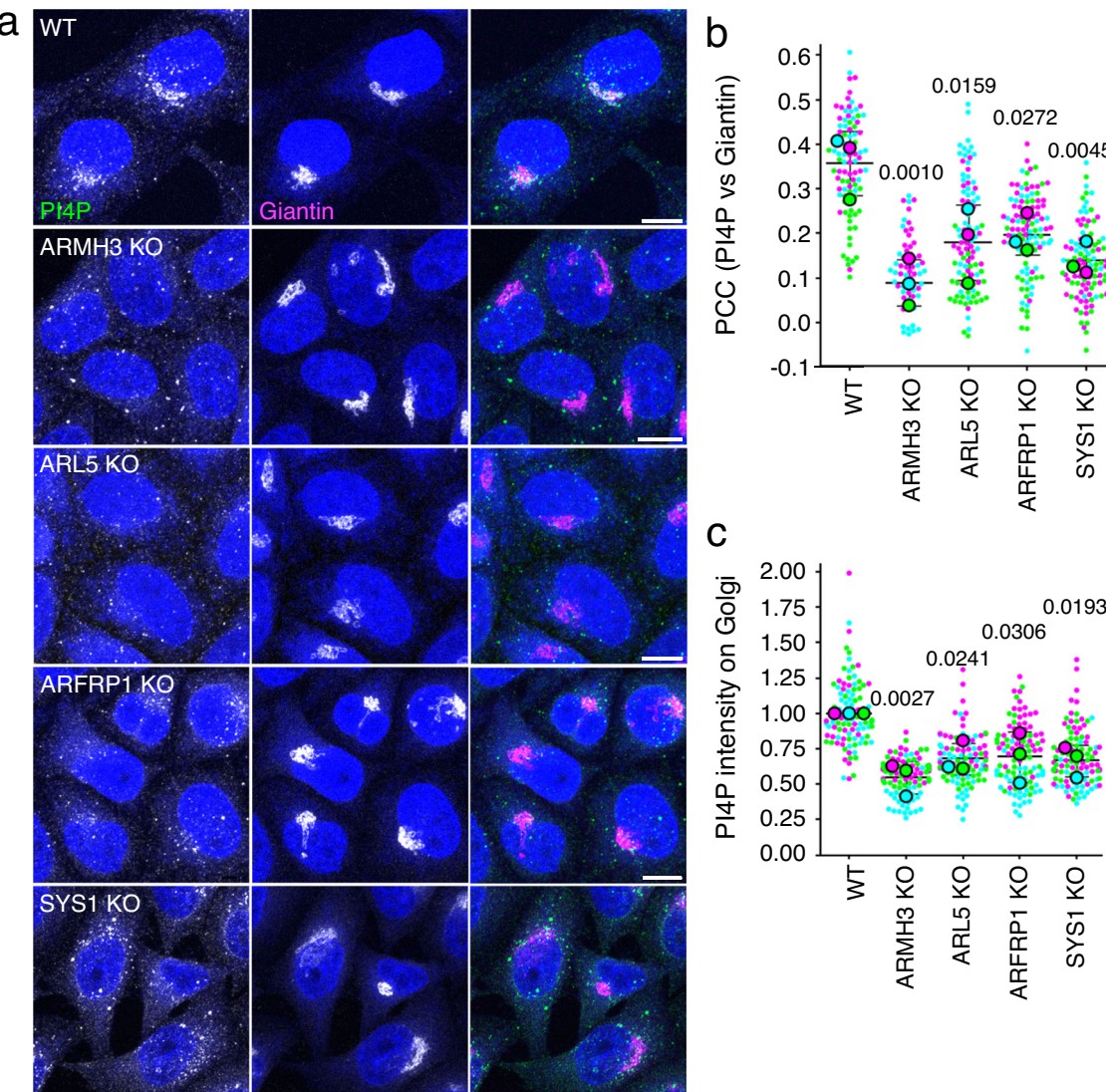

**Fig. 7 | Golgi PI4P is dependent on SYS1, ARFRP1, ARL5, and ARMH3. a** Immunofluorescence microscopy of WT and KO HeLa cells stained for endogenous PI4P (green) and the Golgi protein giantin (magenta). Single-channel images are shown in grayscale with nuclei (DAPI) in blue. Scale bars: 10 μm. Notice the disappearance of PI4P from the Golgi complex and the persistence of vesicular PI4P in all the KO cells. **b** Quantification of the Pearson's correlation coefficient (PCC) of PI4P *vs* giantin from three independent, color-coded experiments such as that shown in panel a. **c** Quantification of PI4P intensity at the Golgi complex relative to the cytosol from experiments such as that shown in panel a. Data are represented as SuperPlots showing the mean ± SD of the means of the individual data points from three independent, color-coded experiments. The statistical significance of the differences was calculated by one-way ANOVA with multiple comparisons using Dunnett's test. *P*-values are indicated on the graphs. Source data are provided in the Source Data file.

further supported by the observations that PI4KB relocalizes to mitochondrially-targeted ARL5B-Q70L with co-expression of GFP-ARMH3, but fails to do so with expression of the ARMH3 mutant[27] that does not interact with PI4KB (Figs. 5a, b and S3).

Although ARL5 and ARMH3 are not required for the recruitment of PI4KB, they are essential for the maintenance of a major pool of PI4P at the Golgi complex. Disruption of the SYS1-ARFRP1-ARL5-ARMH3 axis caused a drastic reduction in Golgi PI4P staining (Figs. 7 and S6a, b). This implies that the presence of PI4KB at the Golgi complex, by binding to ACBD3 or another factor, is insufficient for Golgi PI4P maintenance. Hence, the main role of ARMH3 is the activation of Golgi PI4KB. While the levels of PI4P in the Golgi complex were greatly reduced, PI4P levels in peripheral cytoplasmic puncta were not visibly affected by ARL5 KO or ARMH3 KO. This pool is likely synthesized by other phosphatidylinositol 4-kinases such as PI4K2A

and PI4K2B, which localize at least partially to the endolysosomal system[42,69].

Because ARL5 facilitates retrograde transport from endosomes to the TGN via the GARP complex[6,7], we hypothesized that ARMH3 and PI4KB could also be involved in this process. However, our subsequent experiments showed that ARMH3 and PI4KB did not alter the distribution of three retrograde cargos: TGN46, the CI-MPR, and internalized STxB (Fig. S7a, c). These findings led us to explore other possible functions for ARMH3-PI4KB. Further experiments showed that KO of ARMH3 or PI4KB caused partial dissociation of the PI4P-binding protein GOLPH3, while other PI4P-binding proteins (*e.g.*, AP-1, GGA3) remained unaffected (Fig. S8). These differential effects might be due to the ability of some of these proteins to bind other ligands such as ARF1[53,55–57,70] or utilize the remaining PI4P generated by PI4K2A or PI4K2B[51,54]. GOLPH3's partial dependency on the ARL5-ARMH3-

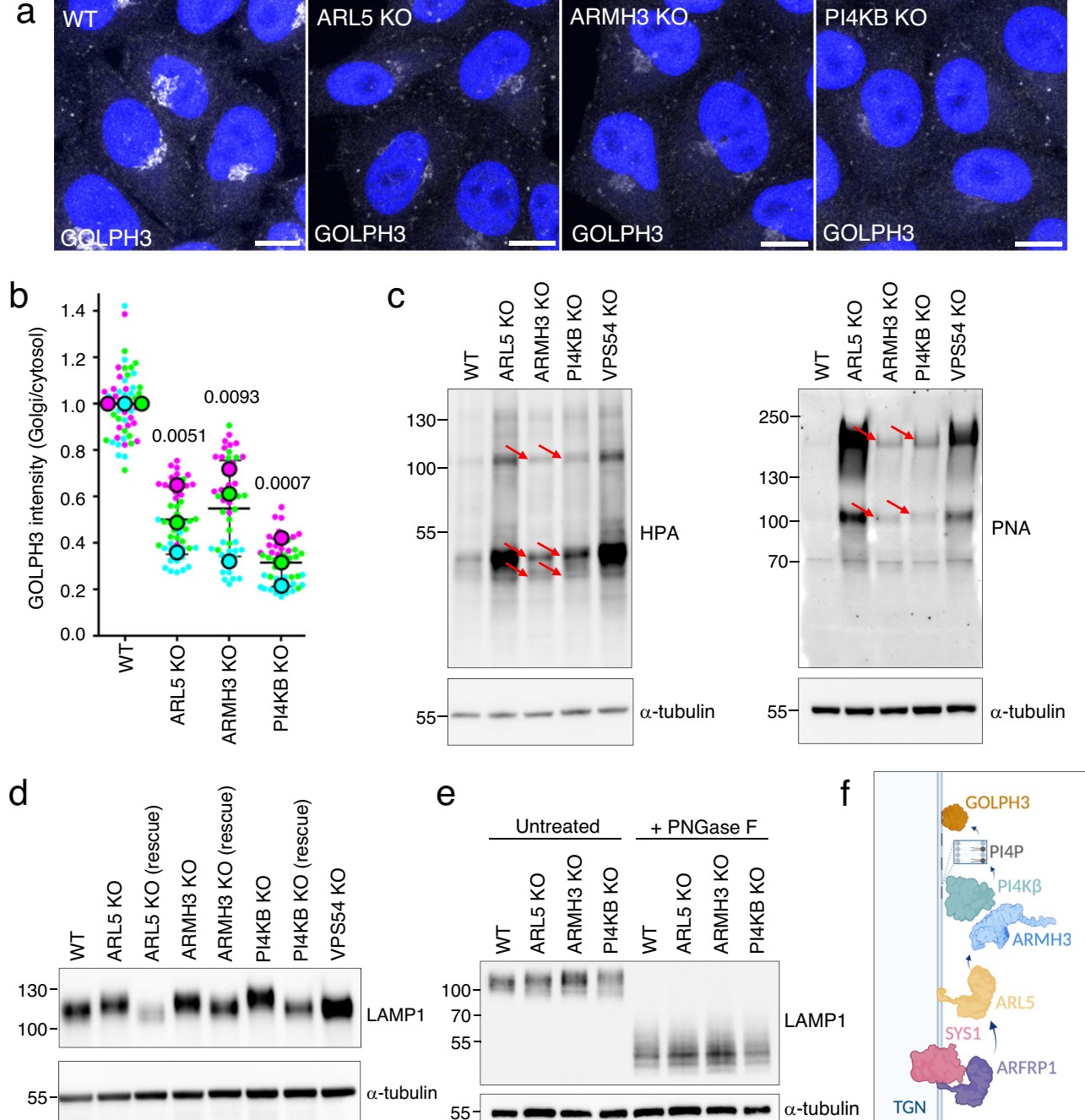

**Fig. 8 | Golgi localization of GOLPH3 and glycan processing is dependent on the ARL5-ARMH3-PI4KB axis. a** WT and KO HeLa cells were immunostained for endogenous GOLPH3. Images are shown in grayscale with nuclei (DAPI) in blue. Scale bars: 10 μm. Notice the decrease in GOLPH3 staining in the KO cells. **b** Quantification of GOLPH3 intensity at the Golgi complex relative to the cytosol from experiments such as that shown in panel a. Data are represented as SuperPlots showing the mean ± SD of the means of the individual data points from three independent, color-coded experiments. The statistical significance of the differences was calculated by one-way ANOVA with multiple comparisons using Dunnett's test. **c** Lectin blotting of WT and KO HeLa cells with HPA-Alexa Fluor 647 and PNA-Fluorescein. Immunoblotting for α-tubulin was used as a loading control. Arrows indicate glycoprotein species whose intensity is changed in the KO cells.

Blots are representative of two independent experiments. **d** Immunoblot analysis of WT, KO, and rescue (ARL5B-mCherry, HA-ARMH3, or Myc-PI4KB stably expressed in corresponding KO cell lines) HeLa cells using antibodies to LAMP1 and α-tubulin (control). Similar results were obtained from two independent biological replicates. **e** WT and KO HeLa cells, untreated or treated with PNGase F, were analyzed by immunoblotting with an antibody to LAMP1. The positions of molecular mass markers (kDa) in panels **c**−**e** are indicated on the left. Blots are representative of three independent experiments. **f** Schematic representation of the role of the SYS1-ARFRP1-ARL5-ARMH3-PI4KB axis in PI4P synthesis at the TGN for recruitment of GOLPH3 to the TGN. Created in BioRender. Bonifacino, J. (2024) BioRender.com/o49h844. Source data are provided in the Source Data file.

PI4KB ensemble for its recruitment to the TGN has important implications for tumorigenesis, as GOLPH3 is a potent oncoprotein[15,64].

Since GOLPH3 mediates COPI-dependent maintenance of a cohort of resident proteins, including glycan-modifying enzymes at the Golgi complex[16–21], we tested for glycoprotein alterations in cells with KO of ARL5, ARMH3, or PI4KB. We found that ARL5 KO caused dramatic changes in the overall *N*- and *O*-linked glycosylation patterns of cellular proteins, similar to those observed by KO of GARP subunits

(Fig. 8c)[7,47]. This is not surprising as GARP is necessary for the fusion of retrograde carriers to the TGN, of which glycosylation enzymes are a cargo. In the absence of GARP, they are unable to escape eventual degradation in the lysosome. ARMH3 KO or PI4KB KO resulted in milder effects, with only a discrete set of proteins displaying altered glycosylation (Fig. 8c). These phenotypes are similar to those previously reported upon KO of PI4KB in Schwann cells in mice[65]. We also specifically found changes in *N*-linked glycosylation of LAMP1 in ARL5-, ARMH3-, and PI4KB-KO cells (Fig. 8d, e). Therefore, the ARL5-ARMH3-PI4KB ensemble promotes terminal glycan modifications of glycoproteins at the Golgi complex.

In conclusion, our findings demonstrate that, like other members of the ARF/ARL family, ARL5 has more than one effector (i.e., at least the GARP complex and ARMH3). Furthermore, each of these effectors is involved in a different function: GARP in retrograde transport and ARMH3 in local PI4KB activation at the TGN. These findings add to the notion that ARF/ARL GTPases play diverse roles, sometimes even at the same cellular location. Our findings also implicate the SYS1-ARFRP1-ARL5-ARMH3 axis as the main driver of PI4P synthesis at the Golgi complex. Although other proteins like ACBD3 can recruit PI4KB[14,36], and other kinases such as PI4K2A can generate PI4P at the Golgi[71], these proteins appear incapable of generating the bulk of Golgi PI4P. ACBD3 plays a major role in recruiting PI4KB for the replication of RNA viruses[36]. Likewise, ARMH3, through its interaction with PI4KB, promotes replication of some RNA[27] and DNA viruses[28]. It would therefore be of interest to investigate whether the SYS1-ARFRP1-ARL5-ARMH3-PI4KB axis contributes to both viral replication and GOLPH3-driven tumorigenesis, potentially offering additional targets for anti-viral and anti-cancer interventions.

## Methods

### Recombinant DNAs

Plasmids encoding human ARL5B (WT, Q70L, or T30N), human ARL1 (Q71L) and human ARFRP1 (WT, Q79L or T31N) cDNAs were described previously[7]. pCI-neo human VPS54-13myc plasmid was described previously[49]. A plasmid encoding human ARL5A cDNA was obtained from Addgene (#67399, submitted by Richard Kahn, Emory University, Atlanta, GA) and its constitutively active and inactive forms (Q70L and T30N) were generated by site-directed mutagenesis using Q5 High-Fidelity DNA Polymerase (New England Biolabs, M0492L). Plasmid pmEGFP-N1 ARL5B (Q70L) was described previously[7]. PCR amplicons of Δ13-ARL5A/B (Q70L or T30N) were subcloned into an MTS-BioID2 containing backbone vector as described previously[12]. PCR amplicons of Δ13-ARL5A/B (WT, Q70L, or T30N), Δ14-ARL1 (Q71L), and Δ14-ARFRP1 (Q79L) were subcloned into pGBK-T7 vector digested with EcoRI-BamHI (Clontech). PCR amplicons of ARL5B (WT, Q70L or T30N) and ARFRP1 (WT, Q79L or T31N) were subcloned into pmCherry-N1 vector digested with EcoRI-BamHI (Clontech). A PCR amplicon of ARL5B (WT or Q70L)-mCherry was subcloned into pQCXIN (Clontech) vector digested with PacI-EcoRI. PCR amplicons of Δ13-ARL5A/B (WT, Q70L or T30N) were subcloned into pGEX-6P-3 digested with BamHI-XhoI (Amersham Pharmacia). The plasmids pEGFP-C1 PI4KB, pEGFP-C1 ARMH3-FLH409AAA[27], mCherry-P4M-SidM[39], and YFP-GalT[43] were described previously. A PCR amplicon of GFP-PI4KB or Myc-PI4KB was subcloned into pQCXIH vector digested with AgeI-PacI (Clontech). PCR amplicons of full-length PI4KB and the N-terminal domain (1-294) of PI4KB were subcloned into pGAD-T7 vector digested with SmaI-XhoI (Clontech).

Due to unexplained toxicity of plasmids containing human ARMH3 when used to transform NEB5α *E. coli* competent cells (New England Biolabs, C2987), the subsequent construction and growth of ARMH3 plasmids was carried out using BL21 (DE3) *E. coli* competent cells (New England Biolabs, C2527I) at 20 °C. A PCR amplicon of human ARMH3 was amplified from HeLa cDNA and subcloned into pmEGFP-1 vector (Addgene, #36409, submitted by Benjamin Glick, University of Chicago) and pcDNA3.1-MTS-BioID2, both of which were digested with EcoRI-BamHI. A PCR amplicon of HA-ARMH3 was subcloned into pQCXIP digested with NotI-EcoRI (Clontech). PCR amplicons of the N-terminal (1-346) and C-terminal (347-689) halves of ARMH3 were subcloned into pmEGFP-1 vector digested with EcoRI-BamHI. PCR amplicons of Δ13-ARL5A/B (Q70L or T30N) were subcloned into the multiple cloning site I (MCSI) of pBridge vector digested with EcoRI-BamHI (Clontech). After the initial subcloning, PCR amplicons of either mEGFP or ARMH3 were further subcloned into the multiple cloning site II (MCSII) of pBridge-Δ13-ARL5A/B (Q70L or T30N) digested with NotI-BglII.

Because the pGAD-T7-ARMH3 construct could not be obtained even from BL21 (DE3) cells, a ligation product prepared by subcloning a PCR amplicon of ARMH3 into EcoRI-BamHI-digested pGAD-T7 vector was directly introduced into AH109 yeast cells, which were used for yeast two-hybrid experiments. The sequence of the pGAD-T7 ARMH3 plasmid was validated by purification from yeast cells using Zymoprep Yeast Plasmid Miniprep II (Zymo Research, D2004), amplified by PCR, and Sanger sequenced.

### Cell culture and transfection

HEK293T (ATCC, CRL-3216) and HeLa (ATCC, CCL-2) cells were maintained in DMEM (Quality Biological, 112-319-101) supplemented with 10% FBS (Corning, 35-011-CV), MycoZap Plus-CL (Lonza, VZA-2012) at 5% $CO_2$ and 37 °C. Lipofectamine 2000 (Invitrogen, 11668019), Lipofectamine 3000 (Invitrogen, L3000015), and/or FuGene HD (Promega, E2311) were used for transfections according to the manufacturers' protocols. Cells were imaged live, fixed, or lysed ~24 h after transfection for subsequent experiments.

### CRISPR/Cas9 knockout

VPS54-KO, ARL1-KO, ARL5-KO, ARFRP1-KO and SYS1-KO HeLa cells were described previously[7]. ARMH3-KO and PI4KB-KO HeLa cells were generated using CRISPR/Cas9[72]. The targeting sequences for ARMH3 (5′-CCCAT GCTCC CTACG ACTTG/TATTG TGAAG CTGTC TATCG-3′) and PI4KB (5′-CCTGC TAAGT GTCAT CACGG/GTGTG GGGTA CACGGA CCACG/AGACT CGGGC AGGGA GCTTA-3′) were cloned separately into pSpCas9 (BB)− 2A-GFP plasmid (Addgene, #48138, submitted by Feng Zhang, Massachusetts Institute of Technology). HeLa cells were transfected with two or three plasmids containing the different targeting sequences for the same gene. GFP-positive cells were isolated by flow cytofluorometry after 24 h and single-cell cloned in 96-well plates. KO was confirmed by immunoblotting using specific antibodies.

### Stable cell lines

HeLa cells stably expressing VPS54-13myc were generated by transfecting pCI-neo VPS54-13myc plasmids into WT HeLa cells, followed by antibiotic selection with 750 µg/ml Geneticin (Gibco, 10131035). After selection, single cell clones were isolated by serial dilution and one clone positive for VPS54-13myc signal at the TGN was identified. To generate HeLa cells stably expressing HA-ARMH3, GFP-PI4KB, and/or ARL5B-Q70L-mCherry, retrovirus particles were prepared by transfecting HEK293T cells with pQCXIP-HA-ARMH3, pQCXIH-GFP-PI4KB, pQCXIH-Myc-PI4KB or pQCXIN-ARL5B-(WT or Q70L)-mCherry and retrovirus-packaging plasmids pCMV-Gag-Pol (Cell Biolabs, RV-111) and pCMV-VSV-G (Cell Biolabs, RV-110) using FuGene HD (Promega, E2311) according to the manufacturer's instructions. Medium was collected 24 h after transfection and centrifuged for 10 min at 1000×*g* to remove debris. HeLa cells were immediately infected with the corresponding virus, and stably transduced cells were selected with 2 µg/ml puromycin (Gibco, A11138-03), 500 µg/ml Geneticin and/or 500 µg/ml hygromycin B (Gibco, 10687010). To establish HeLa cells stably expressing GFP-PI4KB, HA-ARMH3, and ARL5B-Q70L-mCherry, a single clone

exhibiting Golgi localization of GFP-PI4KB was isolated by serial dilution after selection.

## Proximity biotinylation using MitoID

ARL5- or ARMH3-interacting proteins were identified by MitoID[11] with modifications[12]. The ARL5 and ARMH3 MitoID experiments were run separately using three and four experimental replicates per condition, respectively. All samples within each experiment were processed simultaneously. HEK293T cells ($5.4 \times 10^6$) were plated on 15-cm plates (two plates per condition) (Corning, 353025). The next day, cells were transfected with 50 μL Lipofectamine 2000 (Thermo Fisher, 11668019) and 25 μg plasmid encoding MTS-BioID2-ARL5A/B-Q70L/T30N, MTS-BioID2-ARMH3, or MTS-BioID2 (negative control). We prepared two 15-mL tubes with Opti-MEM (Gibco, 31985062): one was mixed with the DNA and the second with Lipofectamine 2000. After a 5-min incubation at room temperature, the contents of the tubes were combined, and the mix incubated at room temperature for an additional 20 min. The Lipofectamine/DNA mixture was added directly to the existing media with no subsequent media changes. At 22 h after transfection, 50 μM biotin (Millipore-Sigma, 47868) was added to each plate (1.5 mL from 1 mM stock). At 24 h after biotin addition, cells were scraped from each plate in 4 mL cold PBS, and 2 plates were combined per sample for a total of 8 mL in 15-mL tubes. The cells were pelleted at $500\,g$ for 5 min at 4 °C. The pellets were washed twice, the first without resuspension and the second with resuspension in 8 mL cold PBS. Cell pellets were stored at −80 °C. Thawed cells were resuspended in 5 mL Buffer A (25 mM Tris-HCl, pH 7.4, 150 mM NaCl, 1 mM EDTA, 1% Triton X-100) supplemented with cOmplete, Mini, EDTA-free Protease Inhibitor (Roche, 1836170) and incubated for 1 h at 4 °C with gentle rotation. The soluble fraction was separated by centrifugation for 20 min at 4 °C, 17,000 g. Pierce NeutrAvidin Agarose slurry (Thermo Scientific, 29201) (500 μL, corresponding to 250 μL beads per sample) was washed in 14 mL Buffer A. The supernatant was then incubated with NeutrAvidin Agarose overnight at 4 °C with gentle rotation. The beads were separated from the lysate by centrifugation for 5 min at $500 \times g$ and 4 °C, and washed twice in 3 mL Buffer B (2% SDS), 3 times in 5 mL Buffer C (50 mM HEPES, 1 mM EDTA, 0.5 M NaCl, 0.1% deoxycholic acid, 1% Triton X-100, pH 7.5), and once in 5 mL Final Wash Buffer (50 mM Tris-HCl, pH 7.4, 50 mM NaCl). Between washes, samples were centrifuged for 5 min at 4 °C, 500 g. Lastly, the washed NeutrAvidin Agarose was resuspended in 75 μL 4 x Laemmli buffer (Bio-Rad, 1610747) and samples were heated for 10 min at 99 °C. Sixty microliters were loaded onto 12% TGX precast gels (Bio-Rad, 4561043), which were run for a few minutes to allow the sample to enter the gel.

## Mass spectrometry

Bands containing the entire sample were excised from the gel. Subsequently, samples were subjected to reduction with 10 mM Tris (2-carboxyethyl) phosphine (TCEP) for 1 h, followed by alkylation with 10 mM NEM for 10 min. Digestion with trypsin was carried out at 37 °C overnight. Peptides were then extracted from the gel and desalted using Oasis HLB μElution plates (Waters, 186001828BA). Each sample's digest was injected into an Ultimate 3000 RSLC nano HPLC system (Thermo Fisher). Peptides were separated on an ES802 column over a 95- or 66-min gradient for the ARL5 MitoID and ARMH3 MitoID, respectively, with mobile phase B (98% acetonitrile, 1.9% $H_2O$, 0.1% formic acid) increased from 3% to 22%. LC-MS/MS data were acquired on an Orbitrap Lumos mass spectrometer (Thermo Fisher) in data-dependent acquisition mode. MS1 scans were performed in the Orbitrap with a resolution of 120 K, covering a mass range of 375–1500 m/z, and an AGC target of $1 \times 106$. Quadrupole isolation was used with a window of 1.6 m/z. MS/MS scans were triggered when the intensity of precursor ions with a charge state between 2 and 6 reached $1 \times 104$. MS2 scans were conducted in the ion trap using the HCD method with a collision energy fixed at 30%. The instrument operated in top speed mode, with an MS1 scan performed every 3 s, acquiring as many MS2 scans as possible within the 3-s cycle. Database search and label-free quantification were carried out using Proteome Discoverer 2.4 software. Up to 2 missed cleavages were allowed for trypsin digestion. NEM on cysteines and oxidation on methionine were set as fixed and variable modifications, respectively. Mass tolerances for MS1 and MS2 scans were set to 10 ppm and 0.6 Da, respectively. The search results were filtered by a false discovery rate of 1% at the protein level. Sequest HT was employed for database search against the Sprot Human database, and peptide spectrum match validation was performed using Percolator. Data were normalized to total peptide abundance. The low abundance resampling method was used for missing value imputation. The summed intensity of unique peptides was used for protein ratio calculation. The maximum and minimum fold changes allowed were set to 1000 and 0.01, respectively. The abundance ratio was not subjected to transformation. An ANOVA with a Tukey post-test was employed for hypothesis testing in Fig. 1c and a two-tailed T-test was employed in Fig. 4b. Proteins with a $\log_2$ fold change ≥1 and a $p$-value of ≤0.05, were considered significantly changed.

## Antibodies and lectins

The following primary antibodies (supplier, catalog number, working dilution) were used for immunoblotting and/or immunofluorescence microscopy: chicken anti-BioID2 (BioFront Technologies, BID2-CP-100, 1:1000 for IF), mouse HRP-conjugated anti-GFP (Miltenyi Biotec, 130-091-833, 1:2000 for WB), rat anti-HA epitope (Roche, 12158167001, 1:1000 for IF, 1:5000 for WB), rabbit anti-GFP (Invitrogen, A-11122, 1:1000 for IF), mouse anti-PI4KB (BD Biosciences, 611817, 1:200 for IF, 1:1000 for WB), mouse HRP-conjugated anti-α-tubulin (Santa Cruz Biotechnology, sc-32293 HRP, 1:5000 for WB), mouse anti-PI4P (Echelon Biosciences, Z-P004, 1:200 for IF), rabbit anti-giantin (Abcam, ab80864, 1:500 for IF), rabbit anti-ARMH3 (Invitrogen, PA5-62264, 1:200 for WB), rabbit anti-GOLPH3 (Abcam, ab98023, 1:200 for IF, 1:1000 for WB), mouse anti-CI-MPR (Abcam, ab2733, 1:200 for IF), sheep anti-TGN46 (Bio-Rad, AHP500G, 1:200 for IF, 1:5000 for WB), mouse anti-γ1-adaptin (BD Biosciences, 610385, 1:200 for IF), rabbit anti-β-COP (Invitrogen, PA1-061, 1:200 for IF), mouse anti-GGA3 (BD Biosciences, 612310, 1:200 for IF), mouse anti-ACBD3 (Sigma-Aldrich, WH0064746M1, 1:200 for IF), goat anti-ST6GAL1 (R&D Systems, AF5924, 1:200 for IF), mouse anti-LAMP1 to detect fully glycosylated protein (DSHB, H4A3-C, 1:5000 for WB) (Fig. 8d), rabbit anti-LAMP1 to detect both glycosylated and deglycosylated protein (Cell Signaling Technology, 9091, 1:1000 for WB) (Fig. 8e), Alexa Fluor 555 goat anti-chicken IgY (Invitrogen, A-21437, 1:1000 for IF), Alexa Fluor 488 donkey anti-rabbit IgG (Invitrogen, A-21206, 1:1000 for IF), Alexa Fluor 555 donkey anti-mouse IgG (Invitrogen, A-31570, 1:1000 for IF), Alexa Fluor 647 donkey anti-rat IgG (Invitrogen, A-48272, 1:1000 for IF), Alexa Fluor 488 donkey anti-mouse IgG (Invitrogen, A-21202, 1:1000 for IF), Alexa Fluor 546 donkey anti-rabbit IgG (Invitrogen, A-10040, 1:1000 for IF). HRP-conjugated donkey anti-sheep (R&D Systems, HAF016, 1:2000 for WB), HRP-conjugated goat anti-rabbit IgG (Jackson Immuno Research, 111-035-003, 1:2000 for WB), HRP-conjugated goat anti-rat IgG (Jackson Immuno Research, 112-035-143, 1:2000 for WB), HRP-conjugated goat anti-mouse IgG (Jackson Immuno Research, 715-035-150, 1:2000 for WB).

Lectin *Helix pomatia* (HPA)-Alexa Fluor 647 (Invitrogen, L32454) and *Arachis Hypogaea* (peanut) Agglutinin (PNA)-Fluorescein from Lectin Kit I, Fluorescein (Vector Laboratories, FLK-2100) were used for the lectin blot assays.

## Immunofluorescent staining

Cells plated on fibronectin-coated cover glasses were fixed with 4% paraformaldehyde in PBS. Cells were permeabilized and blocked with 0.1% saponin (Sigma-Aldrich) and 1% BSA in PBS for 30 min at room

temperature. Primary antibodies were diluted in the same buffer and incubated on cells for 90 min at room temperature. Alexa Fluor secondary antibodies were diluted in the same buffer containing DAPI (Thermo Fisher Scientific, D1306), and cells were incubated for 1 h at room temperature. The coverslips were mounted on glass slides using ProLong Gold Antifade (Invitrogen, P36934).

For immunostaining of endogenous PI4P (Figs. 7 and S6a), a special protocol was used as described previously[40]. In brief, cells on fibronectin-coated cover glass were fixed with 2% paraformaldehyde in PBS, and permeabilized with 20 μM digitonin in buffer A (20 mM PIPES pH 6.8, 137 mM NaCl, 2.7 mM KCl) for 5 min at room temperature. After washing 3 times with buffer A, the coverslips were incubated in blocking buffer (5% FBS in buffer A) for 45 min at room temperature. Primary antibodies diluted in blocking buffer were incubated with cells for 90 min at room temperature. After washing three times with buffer A, the coverslips were incubated with Alexa Fluor secondary antibodies and DAPI diluted in blocking buffer for 60 min at room temperature, and washed three times with buffer A.

## Microscopy

Live cells for imaging were cultured on borosilicate 4-chambered coverglass (Lab-Tek, 155383). Live and fixed cells were imaged by confocal microscopy (LSM780 or LSM880, Carl Zeiss) with an oil-immersion 63×/1.40 NA Plan-Apochromat Oil DIC M27 objective lens (Carl Zeiss), or by structured illumination microscopy (SIM) (Zeiss Elyra PS.1, Carl Zeiss) with a Plan-Apochromat 63×/1.4 NA objective lens at room temperature. For SIM, three orientation angles of the excitation grid with five phases each were acquired for each z plane. Images were then reconstructed with the SIM module in Zeiss ZEN Black software (version 14.0.27.201) using the automatic setting. Z-stack calibration images of a multi-speck bead slide (Carl Zeiss AG, 1783−455) were acquired with equivalent microscope settings, reconstructed as above, and used to correct chromatic aberration by applying affine fit of experimental images to bead calibration images using the channel alignment processing module. Image settings (i.e., gain, laser power, and pinhole) were kept constant for comparison. Images were processed with Fiji (https://imagej.net/software/fiji/)[73], including brightness adjustment, contrast adjustment, channel merging, and cropping.

## Yeast two-hybrid and three-hybrid assays

Yeast two-hybrid assays were performed using pGBKT7 vectors that harbor ARL5, ARL1, or ARFRP1 variants lacking the N-terminal α-helix, pGADT7-ARMH3 or -PI4KB, pVA3 and pTD1. The yeast strain (AH109), selection medium, culture conditions, and transformation protocol were previously described[74]. Yeast three-hybrid assays were performed using pBridge vectors that harbor both ARL5 variants without the N-terminal α-helix and ARMH3 (or GFP as a negative control), pGADT7-PI4KB, pVA3, and pTD1. The yeast strain (HF7c) and selection medium, culture conditions, and transformation protocol were as previously described[75].

## Purification of GST fusion proteins and GST-pulldown

BL21 (DE3) cells expressing GST or GST-Δ13-ARL5A/B (WT, Q70L or T30N) proteins were pre-cultured in 5 mL LA medium (Luria-Bertani broth supplemented with 100 μg/mL ampicillin) at 37 °C with 200 rpm rotation overnight. The cultures were transferred into 150 mL LA medium supplemented with 1 mM isopropyl β-D-1-thiogalactopyranoside and further cultured at 19 °C with 200 rpm rotation overnight. The bacterial cultures were pelleted by centrifugation for 15 min at 4 °C, 4,000 g, and pellets resuspended in 20 mL of 25 mM HEPES pH 7.4, 25 mM KCl, 2 mM MgCl₂, cOmplete EDTA-free Protease Inhibitor Cocktail (Roche, 118735800), 100 μg/mL Dnase-I, 100 μg/mL lysozyme. Following sonication and centrifugation for 30 min at 4 °C, 14,000×g, cleared lysates were incubated with 200 μL glutathione-Sepharose 4B beads (Cytvia, 17-0756-05) at 4 °C overnight, with gentle end-to-end rotation. The beads were washed three times with 20 mL of 25 mM HEPES pH 7.4, 25 mM KCl, 2 mM MgCl₂, resuspended in 100 μL of 25 mM HEPES pH 7.4, 25 mM KCl, 2 mM MgCl₂, cOmplete EDTA-free Protease Inhibitor Cocktail, and stored at 4 °C for subsequent pulldown experiments.

HeLa cells stably expressing HA-ARMH3 and/or GFP-PI4KB were lysed in 0.5% IGEPAL-CA630, 25 mM HEPES pH 7.4, 25 mM KCl, 2 mM MgCl₂, and cOmplete EDTA-free Protease Inhibitor Cocktail, cleared by centrifugation at 14,000 g, and 700 μL of the extract (5,000 cells/μl) was incubated with 40 μL of beads with the GST proteins described above at 4 °C overnight. The beads were washed three times with 700 μL 0.05% IGEPAL-CA630, 25 mM HEPES pH 7.4, 25 mM KCl, 2 mM MgCl₂. Bound proteins were eluted by incubation for 5 min at 98 °C in Laemmli SDS-PAGE sample buffer containing 2.5% 2-mercaptoethanol. The samples were analyzed by SDS-PAGE and immunoblotting.

## Immunoblotting

Unless otherwise specified, lysis buffer composed of 1% Triton X-100, 50 mM Tris-HCl pH 7.4, 150 mM NaCl, and cOmplete EDTA-free Protease Inhibitor Cocktail was used for preparing cell lysates. To detect endogenous ARMH3 (Fig. 7A), cells were lysed in lysis buffer composed of 0.5% IGEPAL-CA630 (Sigma-Aldrich, I8896-50ML), 25 mM HEPES pH 7.4, 25 mM KCl, 2 mM MgCl₂ and cOmplete EDTA-free Protease Inhibitor Cocktail. Cells were trypsinized, collected in 1.5 ml tubes, washed with PBS once, lysed in a corresponding lysis buffer, and cleared by centrifugation at 14,000 g. Laemmli SDS-PAGE sample buffer (Bio-Rad, 161-0747) containing 2.5% 2-mercaptoethanol (LSB) was added to the lysate, and the lysate was incubated for 5 min at 98 °C. Proteins were resolved by SDS-PAGE and subsequently transferred to nitrocellulose membranes. Membranes were blocked for 0.5–1 h with 3% nonfat milk (Bio-Rad, 1706404) in TBS-T (TBS supplemented with 0.05% Tween 20; Sigma-Aldrich, P9416-100ML) before incubation with primary antibody diluted in TBS-T with 3% nonfat milk. Membranes were washed three times for 20 min in TBS-T and incubated for 2–3 h in HRP-conjugated secondary antibody (1:5,000) diluted in TBS-T with 3% nonfat milk. Membranes were washed three times in TBS-T and visualized using Clarity ECL Western Blot substrate (Bio-Rad, 1705061).

## Lectin blotting

Protein samples were prepared as above, resolved by SDS-PAGE, and subsequently transferred to a nitrocellulose membrane. The nitrocellulose membrane was blocked with 3% bovine serum albumin (BSA) in PBS for 30 min. The fluorescent lectins were diluted 1:1,000 in 3% BSA in PBS, incubated with the membrane for 30 min, washed in PBS four times for 4 min each and imaged using a Bio-Rad ChemiDoc system.

## PNGase F treatment

Protein lysates were prepared as above. A fraction of the lysate was treated with PNGase F (New England Biolabs, P0704S) according to the manufacturer's protocol for denaturing reaction conditions. To best visualize mobility shifts of differentially glycosylated LAMP1, independent of its expression levels, less untreated and treated ARL5 and PI4KB KO samples were loaded (50% and 64%, respectively, relative to WT and ARMH3 KO). Loading volume was standardized across all samples by diluting with lysis buffer/LSB.

## STxB uptake

Uptake of Cy3-STxB was performed as previously described[44,76]. Briefly, HeLa cells were incubated for 30 min in STxB uptake medium (DMEM supplemented with 1% BSA). Cells were subsequently incubated with 0.5 μg/ml Cy3-STxB in STxB uptake medium for 15 min and washed with PBS once, before being incubated for 1 h with prewarmed

complete DMEM. Cells were then fixed with 4% paraformaldehyde in PBS and immunostained for giantin as a marker for the Golgi complex.

## Quantification and statistics

Quantification of Pearson's correlation coefficient (PCC) between two channels (Figs. 2b, 5b, 7b, and S3b) was performed using the PSC colocalization plug-in of Fiji. Maximum intensity projections of z-stack images were acquired for each condition, and each cell was selected by drawing a region of interest (ROI) encircling the surface of each cell with the selection brush tool. The PCC was then measured for each cell.

Quantification of yeast growth area in the yeast two-hybrid assay (Fig. 2d) was performed using Fiji. Eight-bit images of day-3 plates were acquired for each condition, and the images were converted to black and white (binary) images by the 'Threshold' function (minimum value was set to 75 and maximum value was set to 255) in Fiji. Yeast spots were encircled with the oval selection tool and RawIntDen values (total intensity in the selected area) were measured using the function 'Measure' in Fiji. Total yeast positive pixels were calculated by dividing RawIntDen value by 255. The growth area of ARMH3 and p53 double-transformants was used as a control, and each growth area in -His plates was divided by that in +His plates.

Quantification of the percentage of cells with GFP-ARMH3 (Fig. 3b, d) on the Golgi was performed by counting >100 cells per experiment per sample. Percentages were calculated with Excel (Microsoft).

Quantification of the abundance of GST-pulled-down proteins in immunoblots (Fig. 4g, i) was performed using the function 'Lane and Bands' in Image Lab (Bio-Rad). The band intensities of HA-ARMH3 or GFP-PI4KB were normalized to the corresponding input, and further normalized to the intensities of GST proteins in each lane.

The intensity of GOLPH3, PI4KB, PI4P, and VPS54-13myc on the Golgi apparatus (Figs. 6c, e, 7c, 8b, S2b, S5c, and S6b) was measured using the function 'Measure' in Fiji. Maximum intensity projections of z-stack images were acquired for each condition, and the region of the Golgi apparatus in each cell was selected by drawing a region of interest (ROI) with the selection brush tool. Part of the cytosol in cells was randomly selected with the same tool, and mean intensities of the targets on the Golgi apparatus and those in the cytosol were measured. The mean intensities of the targets on the Golgi apparatus were divided by the mean intensities of those in the cytosol.

Data are presented as means ± SD or SuperPlots[77]. Statistical significance was calculated using unpaired $t$-test or one-way ANOVA with multiple comparisons using Dunnett's or Tukey's test (Prism 9 for macOS). All graphs were drawn using Prism 9 or Python3 Matplotlib and Seaborn libraries for macOS.

The total number of samples ($n$) analyzed in each experiment is indicated in the figure legends.

No statistical methods were used to predetermine sample sizes, but our sample sizes are like those reported in previous publications. Instead, multiple independent experiments were carried out using several sample replicates as detailed in the figure legends. Data collection and analysis were not performed blind to the conditions of the experiments. Data collection was not randomized. Data distribution was assumed to be normal, but this was not formally tested.

## Reporting summary

Further information on research design is available in the Nature Portfolio Reporting Summary linked to this article.

## Data availability

All data are available in the main text, figures, supplementary materials, or source data files. All raw data, peak list, and search results for proteomic analyses are available on the MassIVE repository (Fig. 1: MSV000094675 [https://massive.ucsd.edu/ProteoSAFe/private-dataset.jsp?task=5b28ac1a4b9f423186e4df468a00dc14], Fig. 4: MSV000094677 [https://massive.ucsd.edu/ProteoSAFe/private-dataset.jsp?task=16f8a7e98aa3438e9b022274f19ef83f] and Proteo-meXchange consortium (Fig. 1: PXD051988, Fig. 4: PXD051993). Reagents generated and microscopy data reported in this study are available upon request and should be directed to the lead contact, Juan S. Bonifacino (juan.bonifacino@nih.gov). Source data are provided with this paper.

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

## Acknowledgements
We thank Benjamin Glick, Richard Kahn, and Feng Zhang for gifts of reagents through Addgene, John Burke for pEGFP-C1 ARMH3-FLH409AAA, and María López-Ocasio and Pradeep Dagur at the NHLBI Flow Cytometry Core for cell sorting. We thank Rahul Raina at the NICHD for valuable advice on buffer composition used for binding experiments. We also thank Xiaolin Zhu for her expert technical assistance. This work was supported by the Intramural Program of NICHD (ZIA HD001607 to J.S.B.). Figure graphics were created with Biorender.com. The funders had no role in study design, data collection and analysis, decision to publish, or preparation of the manuscript.

## Author contributions
T.K.K., M.I., A.E.G., T.B., and J.S.B. designed the study. T.K.K., M.I., and A.E.G. performed most of the experiments. Y.L. performed mass spectrometry analysis. J.S.B. supervised the study. M.I., A.E.G., and J.S.B. wrote the original draft of the manuscript. All the authors reviewed and edited the manuscript before submission.

## Funding

## Competing interests
The authors declare no competing interests.
