## [Peer Review file · Nature Communications]

ARMH3 is an ARL5 effector that promotes PI4KB-catalyzed PI4P synthesis at the *trans*-Golgi network

Corresponding Author: Dr Juan Bonifacino

Version 0:

Reviewer comments:

Reviewer #1

(Remarks to the Author)

In the current manuscript, authors reported identification of a new effector molecule for ARL5, a Golgi resident ARF like small GTPase. Using multiple approaches, including Mito-ID based proteomics, Yeast-2 hybrid assay and GST pull down assay, authors showed that the GTP bound activated form the GTPase directly interacts to ARMH3, also known as C10orf7. They further demonstrated that the activated GTPase could interact with PIK4B via ARMH3. Though Arl5 and ARMH3 was not crucial for PIK4B recruitment on Golgi, they regulated the latter's activity. Accordingly, the depletion of Arl5, ARMH3 led to reduction in GOLPH3 recruitment at TGN causing defect in N/O glycosylation of cellular proteins thus highlighting functional implication of the GTPase-effector interaction. As rightly stated by the authors, the study integrates two already established apparently disconnected molecular processes revealing the role of a hypothetical tri-partite complex in Golgi function.

Overall, the experimental methods utilized to derive the above results are very robust and include appropriate controls. However, largely the results involving sub-cellular localization relied on overexpression of proteins of interest. Authors may want to use the appropriate antibodies to demonstrate multiple co-localization results reported in the manuscript. For example, the MTS-ARL5 co-localization with GFP-ARMH3(Fig. 2) demonstrates the specificity of the interaction but under much higher concentration than the cellular concentration of the GTPase and its effector. Use of α -ARL5 and α -ARMH3 antibody will showcase the co-localization under cellular condition. Additionally, it will be interesting to reproduce the interaction results by co-immunoprecipitation or nanobody based precipitation methods.

Authors have proposed the presence of tripartite ARL5-ARMH3-PIK4B complex. Earlier study (PMID: 31829496) on ARMH3 and PIK4B identified the interface amino acids and through their site-directed mutagenesis, authors have shown that PIK4B is important for ARMH3 recruitment. The results from the current study is consistent and explains why PIK4B recruitment on Golgi is not affected upon ARL5/ARMH3 silencing. The authors may need to discuss why the overexpression of ARL5(QL) led to enhanced PIK4B recruitment in the presence of HA-ARMH3. In this context, exploiting the mutant ARMH3 which is deficient in binding to PIK4B will be of particular use. It would also provide better molecular description about the tripartite complex. In the context of PIK4B activity, authors should discuss the possible explanations for enhancement of PIK4B activity upon ARL5 and/or ARMH3 overexpression.

The authors showed (Fig. 8) that ARL5 regulates GOLPH3 recruitment through its interaction with ARMH3, and thus contributes to glycosylation of cellular proteins. However, ARL5 also recruits GARP which was shown to be important for retention and stabilization of glycosylation proteins at the Golgi complex. As shown in Fig8C, the effect of depletion of ARL5 and GARP component VPS54 on N/O glycosylation are similar while the effect of ARMH3 is milder as it is for PI4KB. It remains unclear, the effect of ARL5 depletion on the observed glycosylation is mediated by both GARP and ARMH3-PI4KB or via two parallel axes. This also raises the possibility of competition between GARP and ARMH3-PI4KB for ARL5 binding site. The relative protein levels and affinity of these two complexes of the GTPase would determine the functional (glycosylation) fate. Interestingly, ARL5 via GARP also regulates retrograde transport of multiple cargo molecules whereas ARL5-ARMH3 is not required for such retrograde transport. Addressing the GARP angle in the current study will certainly provide better mechanistic views and make the study more comprehensive instead of just an integration of already established studies.

Authors also may want to rephrase the sentence— 'To confirm the interaction of ARMH3 with active ARL5, we examined the intracellular localization.....' One cannot confirm interaction by sub-cellular co-localization, the latter being limited by the resolution of the optical microscopy.

Reviewer #2

(Remarks to the Author)

Comments:

The manuscript presents significant findings that advance the understanding of ARL5 effectors and their role in PI4P synthesis and TGN function. The primary objective of this manuscript is to identify additional effectors of ARL5. The authors utilize the MitoID approach to identify ARMH3 as an ARL5 effector. They then characterize the interaction between ARL5 and ARMH3 using various methods, including yeast two-hybrid assays. These experiments are well-executed and provide strong evidence for the interaction. Additionally, the authors demonstrate the role of ARMH3 in PI4KB activation and in maintaining the PI4P pool at the TGN. Finally, they show that ARMH3 may function in GOLPH3 recruitment and glycan modifications at the TGN. Overall, the manuscript is well-written and presents significant findings that contribute to the understanding of ARL5 function and TGN regulation. However, there are some confusions and issues in the results that need to be addressed through further experiments or corrections before acceptance. Addressing the points mentioned below will strengthen the manuscript and better highlight the significance of the findings.

Major Questions:

1. The function of ARL5 and ARMH3 in the activation of PI4KB and PI4P synthesis: Several data suggest that KO of ARMH3 does not affect the expression and localization of PI4KB, and ARL5 KO only slightly reduces PI4KB expression. Only co-overexpression of ARL5 and ARMH3 enhances the activation/expression of PI4KB. These conclusions are preliminary, especially due to the lack of key rescue experiments and sufficient controls. Can ARL5 or ARMH3 KO be rescued by expressing their respective genes, or can ARMH3 rescue the ARL5 KO phenotype? (Figure 6)
2. Rescue of ARMH3 KO by HA-ARMH3 expression and enhancement by ARL5 in PI4KB expression: Can ARMH3 KO be rescued by expressing HA-ARMH3 and enhanced by ARL5 expression in PI4KB expression? (Figure 6D and E)
3. Function and mechanism of ARL5-ARMH3 in recruiting GOLPH3 and glycan modification: Can the KO phenotypes of GOLPH3 recruitment be rescued by expressing ARL5 or ARMH3 in their KO cell lines? (Figures 7 and 8)
4. Validation of observations with more PI4P probes or genetically encoded sensors through live imaging (Figure 7).

Minor Points:

1. The labeling of Figure 5 is confusing (GFP and GFP-PI4KB).

Reviewer #3

(Remarks to the Author)

ARL5, a member of the ARF family of small GTPases, is recruited to the trans-Golgi network (TGN) by ARFRP1 and SYS1, facilitating endosome-TGN fusion via the GARP complex. Ishida et al. used MitoID and different orthogonal interaction assays to identify ARMH3, which binds only to the active form of ARL5 and is recruited to the TGN through the SYS1-ARFRP1-ARL5 pathway. ARMH3 is not essential for cargo transport but activates PI4KB, maintaining phosphatidylinositol 4-phosphate (PI4P) at the TGN. This activation supports GOLPH3 recruitment and glycan modifications at the TGN, identifying the SYS1-ARFRP1-ARL5 axis as a novel regulator of PI4KB-dependent PI4P generation.

The manuscript is well written, shows a clear stepwise investigational path with well-controlled experiments. Potential interactions are confirmed by various orthogonal assays.

Major comments:

No proteomics data are provided. The reviewer login details on pride refer to a different unrelated proteomics dataset. The materials and methods part refers to an excel file which is not available. At the moment it is impossible to assess the proteomics data and the manuscript can therefore not be accepted at this moment. Confirmation of interactions at the endogenous level can further improve the manuscript.

Minor comments:

while MitoID has interesting advantages, it also has some disadvantages. Please provide some context on the method in the manuscript so readers can understand the approach.

Line 109: explain briefly why the N-ter was deleted.

Consider combining figures (e.g. 1 and 2).

Transient expression can result in massive overexpression in HEK293T cells. How does the expression level relate to endogenous levels? Can this affect the findings with MitoID?

Indicate which version of the human SWISSPROT database was used.

Please also refer to the correct proteomics data in the data availability part (PRIDE accession).

Version 1:

Reviewer comments:

Reviewer #1

(Remarks to the Author)

The revised manuscript after addressing most of the comments from the reviewer looks much more complete. I have one suggestion. In one of the authors' cited references on ARL5 (PMID:34798070), the endogenous ARL5 was shown by IF. It may be worth checking the reference for the detailed protocol for IF. Though, it may be due to high level of expression of ARL5 protein in the breast cancer cell line, the IF provided endogenous localiation of ARL5, it is worth checking the protocol

provided in the reference. It is also possible that the authors may have already tried various fixing and permeabilization methods used in the field.

Reviewer #2

(Remarks to the Author)

The authors conducted extensive experiments to address the reviewers' comments and corrected several issues raised in previous reviews. They identified a novel effector of ARL5, ARMH3, and demonstrated its role in PI4KB-catalyzed PI4P synthesis and Golph3 recruitment to the TGN. This discovery represents a significant advancement in understanding ARL5's function and will be of great interest in the field. The manuscript is now worthy of publication in Nature Communications.

Reviewer #3

(Remarks to the Author)

My comments were addressed by the authors

To the reviewers: _____

We thank the reviewers for their time, comments, and suggestions to both strengthen and refine our manuscript. We have successfully completed everything recommended except for one experiment (co-localization/co-coimmunoprecipitation of endogenous proteins), which we attempted from two different angles and was found to be experimentally unfeasible. Below we address each comment and detail our changes.

Reviewer #1

In the current manuscript, authors reported identification of a new effector molecule for ARL5, a Golgi resident ARF like small GTPase. Using multiple approaches, including Mito-ID based proteomics, Yeast-2 hybrid assay and GST pull down assay, authors showed that the GTP bound activated form of the GTPase directly interacts with ARMH3, also known as C10orf7. They further demonstrated that the activated GTPase could interact with PIK4B via ARMH3. Though Arl5 and ARMH3 was not crucial for PIK4B recruitment on Golgi, they regulated the latter's activity. Accordingly, the depletion of Arl5, ARMH3 led to reduction in GOLPH3 recruitment at TGN causing defect in N/O glycosylation of cellular proteins thus highlighting functional implication of the GTPase-effector interaction. As rightly stated by the authors, the study integrates two already established apparently disconnected molecular processes revealing the role of a hypothetical tri-partite complex in Golgi function.

Overall, the experimental methods utilized to derive the above results are very robust and include appropriate controls.

We thank the reviewer for recognizing that our study integrates two seemingly unrelated molecular processes, revealing the role of the tri-partite ARL5-ARMH3-PI4KB complex in Golgi function, and for acknowledging the robustness and appropriate controls of our experimental methods.

1. However, largely the results involving sub-cellular localization relied on overexpression of proteins of interest. Authors may want to use the appropriate antibodies to demonstrate multiple co-localization results reported in the manuscript. For example, the MTS-ARL5 co-localization with GFP-ARMH3 (Fig. 2) demonstrates the specificity of the interaction but under much higher concentration than the cellular concentration of the GTPase and its effector. Use of α -ARL5 and α -ARMH3 antibody will showcase the co-localization under cellular condition. Additionally, it will be interesting to reproduce the interaction results by co-immunoprecipitation or nanobody based precipitation methods.

We agree with the reviewer that the interactions reported in our study would ideally be demonstrated through co-localization and/or co-immunoprecipitation of endogenous proteins. Unfortunately, none of the antibodies we tested for ARL5 and ARMH3 were effective for these applications. These antibodies included:

Antibody name	Company	Catalog #
ARL5A Polyclonal Antibody	Invitrogen	PA5-30509
ARL5A antibody	GeneTex	GTX115803
ARL5B Polyclonal antibody	Proteintech	11694-1-AP
ARL5B Antibody (E-3)	Santa Cruz	sc-393511
ARL5A Antibody (G-9)	Santa Cruz	sc-514680
ARL5A/5B/5C Antibody (D-7)	Santa Cruz	sc-390269
C10orf76 Polyclonal Antibody	Invitrogen	PA5-62812
Anti-ARMH3 antibody (135-185 aa)	St John's Laboratory	STJ194680
C10orf76 Polyclonal Antibody	Invitrogen	PA5-62264

This limitation is not unique to our study, as no other studies on ARL5 or ARMH3 in the literature show data for the endogenous proteins in immunofluorescence or co-immunoprecipitation applications (For ARL5, PMID: 35844135, PMID: 25795912; for ARMH3, PMID: 36921576, PMID: 37195633, PMID: 31829496, PMID: 31519766). This is likely due to the low abundance of the endogenous proteins and the lack of specificity or sensitivity of the available antibodies. Additionally, small GTPase:effector interactions are typically of low affinity, transient, and GTP-dependent, further complicating the detection of these interactions for the endogenous proteins. Indeed, our survey of research articles in the literature showed that no studies of ARL5 showed immunofluorescent staining of the endogenous protein or co-immunoprecipitation with other proteins, and neither did most studies of the broader small GTPase family.

As suggested by the reviewer, we also performed a GFP nanobody pulldown assay, expressing GFP-ARL5A-WT, TN, and QL in HEK293T cells, followed by pulldown with a GFP nanobody in a K⁺-based buffer designed to enhance GTPase binding (see Fig. 4), and immunoblotting for endogenous ARMH3. Conversely, we used the same protocol with GFP-ARMH3 expression and probed for endogenous ARL5. However, we were unable to detect binding in either experiment, likely due to the same issues mentioned earlier: low abundance of the endogenous proteins and lack of antibody specificity and sensitivity.

It is important to note, however, that the original MitolD screen involved the redistribution of endogenous ARMH3 to mitochondria specifically by tagged, GTP-bound ARL5A and ARL5B constructs (Fig. 1). This method benefits from the fact that detection of the endogenous protein was performed via mass spectrometry, which is more sensitive than immunoblotting with the currently available antibodies. Furthermore, we demonstrated the redistribution of endogenous ARL5B to mitochondria by mitochondrially targeted ARMH3 (Fig. 4A).

We also wish to emphasize that, in the case of PI4KB, many of the experiments involved detection of the endogenous protein, as the available antibodies were suitable for both immunofluorescence microscopy and immunoblotting. Thus, whenever possible, we did examine the endogenous proteins.

2. Authors have proposed the presence of tripartite ARL5-ARMH3-PIK4B complex. Earlier study (PMID: 31829496) on ARMH3 and PIK4B identified the interface amino acids and through their site-directed mutagenesis, authors have shown that PIK4B is important for ARMH3 recruitment. The results from the current study is consistent and explains why PIK4B recruitment on Golgi is not affected upon ARL5/ARMH3 silencing. The authors may need to discuss why the overexpression of ARL5(QL) led to enhanced PIK4B recruitment in the presence of HA-ARMH3. In this context, exploiting the mutant ARMH3 which is deficient in binding to PIK4B will be of particular use. It would also provide better molecular description about the tripartite complex. In the context of PIK4B activity, authors should discuss the possible explanations for enhancement of PIK4B activity upon ARL5 and/or ARMH3 overexpression.

We thank the reviewer for suggesting the use of the ARMH3-FLH409AAA mutant, which is unable to bind PI4KB (PMID: 31829496). We used this mutant in two experiments that support our findings. First, we repeated the experiment from Fig. 5, where co-expression of Mito-ARL5B-Q70L and HA-ARMH3 led to GFP-PI4KB re-localization to mitochondria, this time using GFP-ARMH3-FLH409AAA (new Fig. S3). While the ARMH3 mutant was recruited to Mito-ARL5B-Q70L, it failed to re-localize mCh-PI4KB. This supports the conclusion that ARL5 recruits PI4KB to the Golgi complex via interaction with ARMH3. Additionally, we expressed GFP-ARMH3 or GFP-ARMH3-FLH409AAA in ARMH3-KO cells and found no difference in endogenous PI4KB intensity at the Golgi apparatus relative to the cytosol (new Fig. S5b, c). These points are addressed in the Results section (lines 233-235, 261-264) and Discussion (lines 404-407).

3. The authors showed (Fig. 8) that ARL5 regulates GOLPH3 recruitment through its interaction with ARMH3, and thus contributes to glycosylation of cellular proteins. However, ARL5 also recruits GARP which was shown to be important for retention and stabilization of glycosylation proteins at the Golgi complex. As shown in Fig8C, the effect of depletion of ARL5 and GARP component VPS54 on N/O glycosylation are similar while the effect of ARMH3 is milder as it is for PI4KB. It remains unclear, the effect of ARL5 depletion on the observed glycosylation is mediated by both GARP and ARMH3-PIRKB or via two parallel axes. This also raises the possibility of competition between GARP and ARMH3-PI4KB for ARL5 binding site. The relative protein levels and affinity of these two complexes of the GTPase would determine the functional (glycosylation) fate. Interestingly, ARL5 via GARP also regulates retrograde transport of multiple cargo molecules whereas ARL5-ARMH3 is not required for such retrograde transport. Addressing the GARP angle in the current study will certainly provide better mechanistic views and make the study more comprehensive instead of just an integration of already established studies.

We agree with the reviewer that the effect of ARL5 knockout on glycosylation likely reflects the combined effects of both GARP and ARMH3-PI4KB dissociation/inactivation. Since GARP depletion has a much more pronounced impact on glycosylation compared to ARMH3/PI4KB depletion (PMID: 31575603, PMID: 34161137, this study), the glycosylation defect observed in ARL5 knockout cells more closely resembles that of GARP depletion. This explanation is now elaborated in more detail on lines 434-438 of the Discussion section.

As suggested by the reviewer, we also investigated the competition between ARMH3 and GARP for recruitment to the Golgi complex. To test this, we expressed GFP (control) or GFP-ARMH3 in HeLa cells stably expressing VPS54-13myc. We observed a significant decrease in VPS54-13myc intensity at the Golgi relative to the cytosol in cells expressing GFP-ARMH3 (new Fig. S2a, b). This indicates that ARMH3 and GARP compete for binding to ARL5 and supports the idea that ARL5 mediates carbohydrate modifications through two parallel mechanisms. Moreover, this experiment further supports our identification of ARMH3 as a novel ARL5 effector, which was not shown in previous studies. This is detailed in the Results section, lines 192-194.

4. Authors also may want to rephrase the sentence— ‘To confirm the interaction of ARMH3 with active ARL5, we examined the intracellular localization.....’ One cannot confirm interaction by sub-cellular co-localization, the latter being limited by the resolution of the optical microscopy.

We thank the reviewer for pointing out this inaccurate statement. We have changed the wording of this sentence to “To validate a potential interaction between ARMH3 and active ARL5...” on line 136.

Reviewer #2

Comments:

The manuscript presents significant findings that advance the understanding of ARL5 effectors and their role in PI4P synthesis and TGN function. The primary objective of this manuscript is to identify additional effectors of ARL5. The authors utilize the MitolD approach to identify ARMH3 as an ARL5 effector. They then characterize the interaction between ARL5 and ARMH3 using various methods, including yeast two-hybrid assays. These experiments are well-executed and provide strong evidence for the interaction. Additionally, the authors demonstrate the role of ARMH3 in PI4KB activation and in maintaining the PI4P pool at the TGN. Finally, they show that ARMH3 may function in GOLPH3 recruitment and glycan modifications at the TGN. Overall, the manuscript is well-written and presents significant findings that contribute to the understanding of ARL5 function and TGN regulation. However, there are some confusions and issues in the results that need to be addressed through further experiments or corrections before acceptance. Addressing the points mentioned below will strengthen the manuscript and better highlight the significance of the findings.

We thank the reviewer for acknowledging that the "experiments are well-executed and provide strong evidence", and that "the manuscript is well-written and presents significant findings." We also appreciate the reviewer's suggestions to clarify certain issues and further highlight the significance of our findings.

Major Questions:

1. The function of ARL5 and ARMH3 in the activation of PI4KB and PI4P synthesis: Several data suggest that KO of ARMH3 does not affect the expression and localization of PI4KB, and ARL5 KO only slightly reduces PI4KB expression. Only co-overexpression of ARL5 and ARMH3 enhances the activation/expression of PI4KB. These conclusions are preliminary, especially due to the lack of key rescue experiments and sufficient controls. Can ARL5 or ARMH3 KO be rescued by expressing their respective genes, or can ARMH3 rescue the ARL5 KO phenotype? (Figure 6).

We have conducted both these experiments. First, we expressed HA-ARMH3 in ARMH3 KO cells to round out the experiment detailed in Fig. 6d, e. However, based on the result in Fig. 6b, c where PI4KB recruitment to the Golgi complex is not affected in the ARMH3 KO cells, there was not necessarily a phenotype to rescue. Expression of HA-ARMH3 in the ARMH3 KO did not lead to an increase or decrease in PI4KB intensity at the Golgi.

Additionally, we tested whether ARMH3 could increase PI4KB recruitment in ARL5 KO cells by expressing HA-ARMH3. We observed that HA-ARMH3 was cytosolic, as expected, and did not increase PI4KB recruitment to the Golgi in ARL5-KO cells (see image on right for reviewer only), further supporting our model that ARL5 is required for ARMH3 recruitment, which then enhances PI4KB recruitment.

2. Rescue of ARMH3 KO by HA-ARMH3 expression and enhancement by ARL5 in PI4KB expression: Can ARMH3 KO be rescued by expressing HA-ARMH3 and enhanced by ARL5 expression in PI4KB expression? (Figure 6D and E).

The enhancement of PI4KB recruitment by HA-ARMH3 and ARL5B-Q70L-GFP expression in ARL5-KO cells was already shown in Fig. 6d, e. In the new version of the manuscript, we have added another control showing that expression of HA-ARMH3 in the absence of ARL5B-Q70L-GFP does not enhance PI4KB recruitment (new Fig. 6d, e). This further supports our model that both ARL5 and ARMH3 cooperate to recruit PI4KB.

3. Function and mechanism of ARL5-ARMH3 in recruiting GOLPH3 and glycan modification: Can the KO phenotypes of GOLPH3 recruitment be rescued by expressing ARL5 or ARMH3 in their KO cell lines? (Figures 7 and 8).

Yes, we can indeed rescue GOLPH3 recruitment to the Golgi complex in ARL5-, ARMH3-, and PI4KB-KO lines by expressing the corresponding proteins. This is now shown in the new Fig. S9 and detailed in lines 322-324 of the Results section.

4. Validation of observations with more PI4P probes or genetically encoded sensors through live imaging (Figure 7).

To support staining of the endogenous PI4P, we expressed the biosensor mCherry-P4M (PMID: 24711504) in WT, ARL5-KO, ARMH3-KO, and PI4KB-KO HeLa cells and observed a decrease in mCherry-P4M at the Golgi in all KO relative to WT cells (new Fig. S6c) and detailed in lines 280-282 of the Results section.

Minor Points:

1. The labeling of Figure 5 is confusing (GFP and GFP-PI4KB).

This has been corrected.

Reviewer #3 (Remarks to the Author):

ARL5, a member of the ARF family of small GTPases, is recruited to the trans-Golgi network (TGN) by ARFRP1 and SYS1, facilitating endosome-TGN fusion via the GARP complex. Ishida et al. used MitolD and different orthogonal interaction assays to identify ARMH3, which binds only to the active form of ARL5 and is recruited to the TGN through the SYS1-ARFRP1-ARL5 pathway. ARMH3 is not essential for cargo transport but activates PI4KB, maintaining phosphatidylinositol 4-phosphate (PI4P) at the TGN. This activation supports GOLPH3 recruitment and glycan modifications at the TGN, identifying the SYS1-ARFRP1-ARL5 axis as a novel regulator of PI4KB-dependent PI4P generation. The manuscript is well written, shows a clear stepwise investigational path with well-controlled experiments. Potential interactions are confirmed by various orthogonal assays.

We thank this reviewer for acknowledging that our manuscript “is well written, shows a clear stepwise investigational path with well-controlled experiments.”

Major comments:

1. No proteomics data are provided. The reviewer login details on pride refer to a different unrelated proteomics dataset. The materials and methods part refers to an excel file which is not available. At the moment it is impossible to assess the proteomics data and the manuscript can therefore not be accepted at this moment.

We have confirmed that our data is currently accessible on the MASSIVE repository to reviewers using the login information provided in the Reporting Summary and below. The data will be visible on the ProteomeXChange consortium once made public. We did not previously have Excel files with our proteomics data linked to the Results section and understand that, like Reviewer #3, the audience may appreciate

quicker access to the proteomics data than retrieving it from the repository or consortium. We have now added Supplementary Data File 1 (ARL5 MitolD) (lines 117-118) and Supplementary Data File 2 (ARMH3 MitolD) (line 202), and detail this more clearly in the Data Availability section (lines 786-792).

Fig. 1: <https://massive.ucsd.edu/ProteoSAFe/private-dataset.jsp?task=5b28ac1a4b9f423186e4df468a00dc14>

Reviewer login: <ftp://MSV000094675@massive.ucsd.edu>
Password: 15LFQ_ARL5AB

Fig. 4: <https://massive.ucsd.edu/ProteoSAFe/private-dataset.jsp?task=16f8a7e98aa3438e9b022274f19ef83f>

Reviewer Login: <ftp://MSV000094677@massive.ucsd.edu>
Password: 8LFQ_ARMH3

2. Confirmation of interactions at the endogenous level can further improve the manuscript.

See response to Reviewer #1, point 1.

Minor comments:

3. While MitolD has interesting advantages, it also has some disadvantages. Please provide some context on the method in the manuscript so readers can understand the approach.

We value this comment and have provided more context to this method in lines 104-107: "MitolD involves targeting a bait protein to mitochondria, leading to the recruitment and biotinylation of potential interactors. This technique enriches for true soluble and peripheral-membrane protein interactors but does limit the identification of transmembrane interactors.

4. Line109: explain briefly why the N-ter was deleted. Consider combining figures (e.g. 1 and 2).

We addressed the purpose of deleting the N-terminal helix on line 111 and considered combining Figures 1 and 2, but decided to leave it as is to minimize clutter.

5. Transient expression can result in massive overexpression in HEK293T cells. How does the expression level relate to endogenous levels? Can this affect the findings with MitolD?

We agree with the reviewer that transient expression of our constructs for MitolD will lead to overexpression. However, the purpose of these experiments in this

manuscript was to identify potential interactors via a discovery, unbiased approach, with the intent to follow up with in vivo and in vitro studies on prospective candidates. In other words, we did not take the “hits” for granted and used it as a launching pad for identifying novel interactors.

6. Indicate SWISSPROT version.

Sequest HT was employed for database search against the Sprot Human database, and peptide spectrum match validation was performed using Percolator (see lines 595-596).

7. Please also refer to the correct proteomics data in the data availability part (PRIDE accession).

Our data are available on the MassIVE repository, not PRIDE. See response to Reviewer #3, point 1.

To the reviewers:

We thank the reviewers for their time, comments, and suggestions to both strengthen and refine our manuscript. We have successfully completed everything recommended except for one experiment (co-localization/co-coimmunoprecipitation of endogenous proteins), which we attempted from two different angles and was found to be experimentally unfeasible. Below we address each comment and detail our changes.

Reviewer #1

In the current manuscript, authors reported identification of a new effector molecule for ARL5, a Golgi resident ARF like small GTPase. Using multiple approaches, including Mito-ID based proteomics, Yeast-2 hybrid assay and GST pull down assay, authors showed that the GTP bound activated form the GTPase directly interacts to ARMH3, also known as C10orf7. They further demonstrated that the activated GTPase could interact with PIK4B via ARMH3. Though Arl5 and ARMH3 was not crucial for PIK4B recruitment on Golgi, they regulated the latter's activity. Accordingly, the depletion of Arl5, ARMH3 led to reduction in GOLPH3 recruitment at TGN causing defect in N/O glycosylation of cellular proteins thus highlighting functional implication of the GTPase-effector interaction. As rightly stated by the authors, the study integrates two already established apparently disconnected molecular processes revealing the role of a hypothetical tri-partite complex in Golgi function.

Overall, the experimental methods utilized to derive the above results are very robust and include appropriate controls.

We thank the reviewer for recognizing that our study integrates two seemingly unrelated molecular processes, revealing the role of the tri-partite ARL5-ARMH3-PI4KB complex in Golgi function, and for acknowledging the robustness and appropriate controls of our experimental methods.

1. However, largely the results involving sub-cellular localization relied on overexpression of proteins of interest. Authors may want to use the appropriate antibodies to demonstrate multiple co-localization results reported in the manuscript. For example, the MTS-ARL5 co-localization with GFP-ARMH3 (Fig. 2) demonstrates the specificity of the interaction but under much higher concentration than the cellular concentration of the GTPase and its effector. Use of α -ARL5 and α -ARMH3 antibody will showcase the co-localization under cellular condition. Additionally, it will be interesting to reproduce the interaction results by co-immunoprecipitation or nanobody based precipitation methods.

We agree with the reviewer that the interactions reported in our study would ideally be demonstrated through co-localization and/or co-immunoprecipitation of endogenous proteins. Unfortunately, none of the antibodies we tested for ARL5 and ARMH3 were effective for these applications. These antibodies included:

Antibody name	Company	Catalog #
ARL5A Polyclonal Antibody	Invitrogen	PA5-30509
ARL5A antibody	GeneTex	GTX115803
ARL5B Polyclonal antibody	Proteintech	11694-1-AP
ARL5B Antibody (E-3)	Santa Cruz	sc-393511
ARL5A Antibody (G-9)	Santa Cruz	sc-514680
ARL5A/5B/5C Antibody (D-7)	Santa Cruz	sc-390269
C10orf76 Polyclonal Antibody	Invitrogen	PA5-62812
Anti-ARMH3 antibody (135-185 aa)	St John's Laboratory	STJ194680
C10orf76 Polyclonal Antibody	Invitrogen	PA5-62264

This limitation is not unique to our study, as no other studies on ARL5 or ARMH3 in the literature show data for the endogenous proteins in immunofluorescence or co-immunoprecipitation applications (For ARL5, PMID: 35844135, PMID: 25795912; for ARMH3, PMID: 36921576, PMID: 37195633, PMID: 31829496, PMID: 31519766). This is likely due to the low abundance of the endogenous proteins and the lack of specificity or sensitivity of the available antibodies. Additionally, small GTPase:effector interactions are typically of low affinity, transient, and GTP-dependent, further complicating the detection of these interactions for the endogenous proteins. Indeed, our survey of research articles in the literature showed that no studies of ARL5 showed immunofluorescent staining of the endogenous protein or co-immunoprecipitation with other proteins, and neither did most studies of the broader small GTPase family.

As suggested by the reviewer, we also performed a GFP nanobody pulldown assay, expressing GFP-ARL5A-WT, TN, and QL in HEK293T cells, followed by pulldown with a GFP nanobody in a K⁺-based buffer designed to enhance GTPase binding (see Fig. 4), and immunoblotting for endogenous ARMH3. Conversely, we used the same protocol with GFP-ARMH3 expression and probed for endogenous ARL5. However, we were unable to detect binding in either experiment, likely due to the same issues mentioned earlier: low abundance of the endogenous proteins and lack of antibody specificity and sensitivity.

It is important to note, however, that the original MitolD screen involved the redistribution of endogenous ARMH3 to mitochondria specifically by tagged, GTP-bound ARL5A and ARL5B constructs (Fig. 1). This method benefits from the fact that detection of the endogenous protein was performed via mass spectrometry, which is more sensitive than immunoblotting with the currently available antibodies. Furthermore, we demonstrated the redistribution of endogenous ARL5B to mitochondria by mitochondrially targeted ARMH3 (Fig. 4A).

We also wish to emphasize that, in the case of PI4KB, many of the experiments involved detection of the endogenous protein, as the available antibodies were suitable for both immunofluorescence microscopy and immunoblotting. Thus, whenever possible, we did examine the endogenous proteins.

2. Authors have proposed the presence of tripartite ARL5-ARMH3-PIK4B complex. Earlier study (PMID: 31829496) on ARMH3 and PIK4B identified the interface amino acids and through their site-directed mutagenesis, authors have shown that PIK4B is important for ARMH3 recruitment. The results from the current study is consistent and explains why PIK4B recruitment on Golgi is not affected upon ARL5/ARMH3 silencing. The authors may need to discuss why the overexpression of ARL5(QL) led to enhanced PIK4B recruitment in the presence of HA-ARMH3. In this context, exploiting the mutant ARMH3 which is deficient in binding to PIK4B will be of particular use. It would also provide better molecular description about the tripartite complex. In the context of PIK4B activity, authors should discuss the possible explanations for enhancement of PIK4B activity upon ARL5 and/or ARMH3 overexpression.

We thank the reviewer for suggesting the use of the ARMH3-FLH409AAA mutant, which is unable to bind PI4KB (PMID: 31829496). We used this mutant in two experiments that support our findings. First, we repeated the experiment from Fig. 5, where co-expression of Mito-ARL5B-Q70L and HA-ARMH3 led to GFP-PI4KB re-localization to mitochondria, this time using GFP-ARMH3-FLH409AAA (new Fig. S3). While the ARMH3 mutant was recruited to Mito-ARL5B-Q70L, it failed to re-localize mCh-PI4KB. This supports the conclusion that ARL5 recruits PI4KB to the Golgi complex via interaction with ARMH3. Additionally, we expressed GFP-ARMH3 or GFP-ARMH3-FLH409AAA in ARMH3-KO cells and found no difference in endogenous PI4KB intensity at the Golgi apparatus relative to the cytosol (new Fig. S5b, c). These points are addressed in the Results section (lines 233-235, 261-264) and Discussion (lines 404-407).

3. The authors showed (Fig. 8) that ARL5 regulates GOLPH3 recruitment through its interaction with ARMH3, and thus contributes to glycosylation of cellular proteins. However, ARL5 also recruits GARP which was shown to be important for retention and stabilization of glycosylation proteins at the Golgi complex. As shown in Fig8C, the effect of depletion of ARL5 and GARP component VPS54 on N/O glycosylation are similar while the effect of ARMH3 is milder as it is for PI4KB. It remains unclear, the effect of ARL5 depletion on the observed glycosylation is mediated by both GARP and ARMH3-PIRKB or via two parallel axes. This also raises the possibility of competition between GARP and ARMH3-PI4KB for ARL5 binding site. The relative protein levels and affinity of these two complexes of the GTPase would determine the functional (glycosylation) fate. Interestingly, ARL5 via GARP also regulates retrograde transport of multiple cargo molecules whereas ARL5-ARMH3 is not required for such retrograde transport. Addressing the GARP angle in the current study will certainly provide better mechanistic views and make the study more comprehensive instead of just an integration of already established studies.

We agree with the reviewer that the effect of ARL5 knockout on glycosylation likely reflects the combined effects of both GARP and ARMH3-PI4KB dissociation/inactivation. Since GARP depletion has a much more pronounced impact on glycosylation compared to ARMH3/PI4KB depletion (PMID: 31575603, PMID: 34161137, this study), the glycosylation defect observed in ARL5 knockout cells more closely resembles that of GARP depletion. This explanation is now elaborated in more detail on lines 434-438 of the Discussion section.

As suggested by the reviewer, we also investigated the competition between ARMH3 and GARP for recruitment to the Golgi complex. To test this, we expressed GFP (control) or GFP-ARMH3 in HeLa cells stably expressing VPS54-13myc. We observed a significant decrease in VPS54-13myc intensity at the Golgi relative to the cytosol in cells expressing GFP-ARMH3 (new Fig. S2a, b). This indicates that ARMH3 and GARP compete for binding to ARL5 and supports the idea that ARL5 mediates carbohydrate modifications through two parallel mechanisms. Moreover, this experiment further supports our identification of ARMH3 as a novel ARL5 effector, which was not shown in previous studies. This is detailed in the Results section, lines 192-194.

4. Authors also may want to rephrase the sentence– ‘To confirm the interaction of ARMH3 with active ARL5, we examined the intracellular localization.....’ One cannot confirm interaction by sub-cellular co-localization, the latter being limited by the resolution of the optical microscopy.

We thank the reviewer for pointing out this inaccurate statement. We have changed the wording of this sentence to “To validate a potential interaction between ARMH3 and active ARL5...” on line 136.

Reviewer #2

Comments:

The manuscript presents significant findings that advance the understanding of ARL5 effectors and their role in PI4P synthesis and TGN function. The primary objective of this manuscript is to identify additional effectors of ARL5. The authors utilize the MitolD approach to identify ARMH3 as an ARL5 effector. They then characterize the interaction between ARL5 and ARMH3 using various methods, including yeast two-hybrid assays. These experiments are well-executed and provide strong evidence for the interaction. Additionally, the authors demonstrate the role of ARMH3 in PI4KB activation and in maintaining the PI4P pool at the TGN. Finally, they show that ARMH3 may function in GOLPH3 recruitment and glycan modifications at the TGN. Overall, the manuscript is well-written and presents significant findings that contribute to the understanding of ARL5 function and TGN regulation. However, there are some confusions and issues in the results that need to be addressed through further experiments or corrections before acceptance. Addressing the points mentioned below will strengthen the manuscript and better highlight the significance of the findings.

We thank the reviewer for acknowledging that the "experiments are well-executed and provide strong evidence", and that "the manuscript is well-written and presents significant findings." We also appreciate the reviewer's suggestions to clarify certain issues and further highlight the significance of our findings.

Major Questions:

1. The function of ARL5 and ARMH3 in the activation of PI4KB and PI4P synthesis: Several data suggest that KO of ARMH3 does not affect the expression and localization of PI4KB, and ARL5 KO only slightly reduces PI4KB expression. Only co-overexpression of ARL5 and ARMH3 enhances the activation/expression of PI4KB. These conclusions are preliminary, especially due to the lack of key rescue experiments and sufficient controls. Can ARL5 or ARMH3 KO be rescued by expressing their respective genes, or can ARMH3 rescue the ARL5 KO phenotype? (Figure 6).

We have conducted both these experiments. First, we expressed HA-ARMH3 in ARMH3 KO cells to round out the experiment detailed in Fig. 6d, e. However, based on the result in Fig. 6b, c where PI4KB recruitment to the Golgi complex is not affected in the ARMH3 KO cells, there was not necessarily a phenotype to rescue. Expression of HA-ARMH3 in the ARMH3 KO did not lead to an increase or decrease in PI4KB intensity at the Golgi.

Additionally, we tested whether ARMH3 could increase PI4KB recruitment in ARL5 KO cells by expressing HA-ARMH3. We observed that HA-ARMH3 was cytosolic, as expected, and did not increase PI4KB recruitment to the Golgi in ARL5-KO cells (see image on right for reviewer only), further supporting our model that ARL5 is required for ARMH3 recruitment, which then enhances PI4KB recruitment.

2. Rescue of ARMH3 KO by HA-ARMH3 expression and enhancement by ARL5 in PI4KB expression: Can ARMH3 KO be rescued by expressing HA-ARMH3 and enhanced by ARL5 expression in PI4KB expression? (Figure 6D and E).

The enhancement of PI4KB recruitment by HA-ARMH3 and ARL5B-Q70L-GFP expression in ARL5-KO cells was already shown in Fig. 6d, e. In the new version of the manuscript, we have added another control showing that expression of HA-ARMH3 in the absence of ARL5B-Q70L-GFP does not enhance PI4KB recruitment (new Fig. 6d, e). This further supports our model that both ARL5 and ARMH3 cooperate to recruit PI4KB.

3. Function and mechanism of ARL5-ARMH3 in recruiting GOLPH3 and glycan modification: Can the KO phenotypes of GOLPH3 recruitment be rescued by expressing ARL5 or ARMH3 in their KO cell lines? (Figures 7 and 8).

Yes, we can indeed rescue GOLPH3 recruitment to the Golgi complex in ARL5-, ARMH3-, and PI4KB-KO lines by expressing the corresponding proteins. This is now shown in the new Fig. S9 and detailed in lines 322-324 of the Results section.

4. Validation of observations with more PI4P probes or genetically encoded sensors through live imaging (Figure 7).

To support staining of the endogenous PI4P, we expressed the biosensor mCherry-P4M (PMID: 24711504) in WT, ARL5-KO, ARMH3-KO, and PI4KB-KO HeLa cells and observed a decrease in mCherry-P4M at the Golgi in all KO relative to WT cells (new Fig. S6c) and detailed in lines 280-282 of the Results section.

Minor Points:

1. The labeling of Figure 5 is confusing (GFP and GFP-PI4KB).

This has been corrected.

Reviewer #3 (Remarks to the Author):

ARL5, a member of the ARF family of small GTPases, is recruited to the trans-Golgi network (TGN) by ARFRP1 and SYS1, facilitating endosome-TGN fusion via the GARP complex. Ishida et al. used MitolD and different orthogonal interaction assays to identify ARMH3, which binds only to the active form of ARL5 and is recruited to the TGN through the SYS1-ARFRP1-ARL5 pathway. ARMH3 is not essential for cargo transport but activates PI4KB, maintaining phosphatidylinositol 4-phosphate (PI4P) at the TGN. This activation supports GOLPH3 recruitment and glycan modifications at the TGN, identifying the SYS1-ARFRP1-ARL5 axis as a novel regulator of PI4KB-dependent PI4P generation. The manuscript is well written, shows a clear stepwise investigational path with well-controlled experiments. Potential interactions are confirmed by various orthogonal assays.

We thank this reviewer for acknowledging that our manuscript “is well written, shows a clear stepwise investigational path with well-controlled experiments.”

Major comments:

1. No proteomics data are provided. The reviewer login details on pride refer to a different unrelated proteomics dataset. The materials and methods part refers to an excel file which is not available. At the moment it is impossible to assess the proteomics data and the manuscript can therefore not be accepted at this moment.

We have confirmed that our data is currently accessible on the MASSIVE repository to reviewers using the login information provided in the Reporting Summary and below. The data will be visible on the ProteomeXChange consortium once made public. We did not previously have Excel files with our proteomics data linked to the Results section and understand that, like Reviewer #3, the audience may appreciate

quicker access to the proteomics data than retrieving it from the repository or consortium. We have now added Supplementary Data File 1 (ARL5 MitolD) (lines 117-118) and Supplementary Data File 2 (ARMH3 MitolD) (line 202), and detail this more clearly in the Data Availability section (lines 786-792).

Fig. 1: <https://massive.ucsd.edu/ProteoSAFe/private-dataset.jsp?task=5b28ac1a4b9f423186e4df468a00dc14>

Reviewer login: <ftp://MSV000094675@massive.ucsd.edu>
Password: 15LFQ_ARL5AB

Fig. 4: <https://massive.ucsd.edu/ProteoSAFe/private-dataset.jsp?task=16f8a7e98aa3438e9b022274f19ef83f>

Reviewer Login: <ftp://MSV000094677@massive.ucsd.edu>
Password: 8LFQ_ARMH3

2. Confirmation of interactions at the endogenous level can further improve the manuscript.

See response to Reviewer #1, point 1.

Minor comments:

3. While MitolD has interesting advantages, it also has some disadvantages. Please provide some context on the method in the manuscript so readers can understand the approach.

We value this comment and have provided more context to this method in lines 104-107: "MitolD involves targeting a bait protein to mitochondria, leading to the recruitment and biotinylation of potential interactors. This technique enriches for true soluble and peripheral-membrane protein interactors but does limit the identification of transmembrane interactors.

4. Line109: explain briefly why the N-ter was deleted. Consider combining figures (e.g. 1 and 2).

We addressed the purpose of deleting the N-terminal helix on line 111 and considered combining Figures 1 and 2, but decided to leave it as is to minimize clutter.

5. Transient expression can result in massive overexpression in HEK293T cells. How does the expression level relate to endogenous levels? Can this affect the findings with MitolD?

We agree with the reviewer that transient expression of our constructs for MitolD will lead to overexpression. However, the purpose of these experiments in this

manuscript was to identify potential interactors via a discovery, unbiased approach, with the intent to follow up with in vivo and in vitro studies on prospective candidates. In other words, we did not take the “hits” for granted and used it as a launching pad for identifying novel interactors.

6. Indicate SWISSPROT version.

Sequest HT was employed for database search against the Sprot Human database, and peptide spectrum match validation was performed using Percolator (see lines 595-596).

7. Please also refer to the correct proteomics data in the data availability part (PRIDE accession).

Our data are available on the MassIVE repository, not PRIDE. See response to Reviewer #3, point 1.

Remaining Comments from Reviewers

Reviewer #1 (Remarks to the Author):

The revised manuscript after addressing most of the comments from the reviewer looks much more complete. I have one suggestion. In one of the authors' cited references on ARL5 (PMID:34798070), the endogenous ARL5 was shown by IF. It may be worth checking the reference for the detailed protocol for IF. Though, it may be due to high level of expression of ARL5 protein in the breast cancer cell line, the IF provided endogenous localiation of ARL5, it is worth checking the protocol provided in the reference. It is also possible that the authors may have already tried vareous fixing and pearmeablization methods used in the field.

We confirmed that the ARL5 antibody used in PMID:34798070 was mouse anti-ARL5B (sc393511, Santa Cruz) detailed in our antibody table above. We tested this antibody using similar fixing and permeabilization methods in our cells and, unfortunately, did not observe sufficient staining.

Reviewer #2 (Remarks to the Author):

The authors conducted extensive experiments to address the reviewers' comments and corrected several issues raised in previous reviews. They identified a novel effector of ARL5, ARMH3, and demonstrated its role in PI4KB-catalyzed PI4P synthesis and Golph3 recruitment to the TGN. This discovery represents a significant advancement in understanding ARL5's function and will be of great interest in the field. The manuscript is now worthy of publication in Nature Communications.

We thank Reviewer #2 for their mutual excitement for these findings!

Reviewer #3 (Remarks to the Author):

My comments were addressed by the authors